# Short-term reward experience biases inference despite dissociable neural correlates

Adrian G. Fischer [ID] [1,2], Sacha Bourgeois-Gironde [ID] [3,4] & Markus Ullsperger [ID] [1,2]

Optimal decision-making employs short-term rewards and abstract long-term information based on which of these is deemed relevant. Employing short- vs. long-term information is associated with different learning mechanisms, yet neural evidence showing that these two are dissociable is lacking. Here we demonstrate that long-term, inference-based beliefs are biased by short-term reward experiences and that dissociable brain regions facilitate both types of learning. Long-term inferences are associated with dorsal striatal and frontopolar cortex activity, while short-term rewards engage the ventral striatum. Stronger concurrent representation of reward signals by mediodorsal striatum and frontopolar cortex correlates with less biased, more optimal individual long-term inference. Moreover, dynamic modulation of activity in a cortical cognitive control network and the medial striatum is associated with trial-by-trial control of biases in belief updating. This suggests that counteracting the processing of optimally to-be-ignored short-term rewards and cortical suppression of associated reward-signals, determines long-term learning success and failure.

[1] Otto-von-Guericke University, Institute of Psychology, D-39106 Magdeburg, Germany. [2] Center for Behavioral Brain Sciences, D-39106 Magdeburg, Germany. [3] Université Paris 2 - LEMMA, F-75006 Paris, France. [4] Ecole Normale Supérieure - Institut Jean-Nicod, F-75005 Paris, France. Correspondence and requests for materials should be addressed to A.G.F. (email: adrian.fischer@ovgu.de) or to M.U. (email: markus.ullsperger@ovgu.de)

Mammalian brains are well-accustomed to learning how to maximise reward based on the immediate evaluation of potential outcomes. Such learning by reward experience is model-free, fast, computationally simple, and can be successfully employed in many ecological environments[1,2]. Yet in modern times, we are often provided with information about desirable long-term outcomes, regardless of the immediately rewarding property of an experience. These long-term consequences are sometimes too distant to be experienced at all, but still they influence us. Learning to infer the long-term consequences of actions when immediate rewards are bad indicators of future successes, requires inference based on a model. How humans accomplish this inference, and which neural processes transform potentially biased information regarding short-term rewards into the formation of beliefs about long-term goals, is only beginning to be investigated. For example, people regularly consume fast-food—despite abundant information on the negative long-term consequences—as it feels pleasurable in the short term[3]. This behaviour, in turn, may bias their estimates of the consequences of healthy eating in general. The relative

influence of information over reward on learning leading to subjective decisions is at the heart of problems such as the global obesity epidemic in children and adults[4], but extends to drug- and many other forms of addiction and disorders[5,6].

Cognitive neuroscientists have only recently begun to investigate the neural processes that enable humans to form inferences by employing models of the world and derive actions from these inference-based beliefs. Previous studies have used computational models of decision-making processes to show that model-free and model-based learning are behaviourally dissociable, yet share common neural substrates[7,8]. In addition, the definition of what exactly constitutes a model, is broad[9], and results may strongly depend on the model used to solve the task. In most previous studies, participants learned models of the complex contingencies of context, choices and outcomes, such that the model itself had to be established and successively employed to solve the task. Here, we provide participants with full knowledge about the model itself, such that long-term inferential belief formation is feasible without learning. By manipulating congruence between short-term rewards and inference, we study how reward

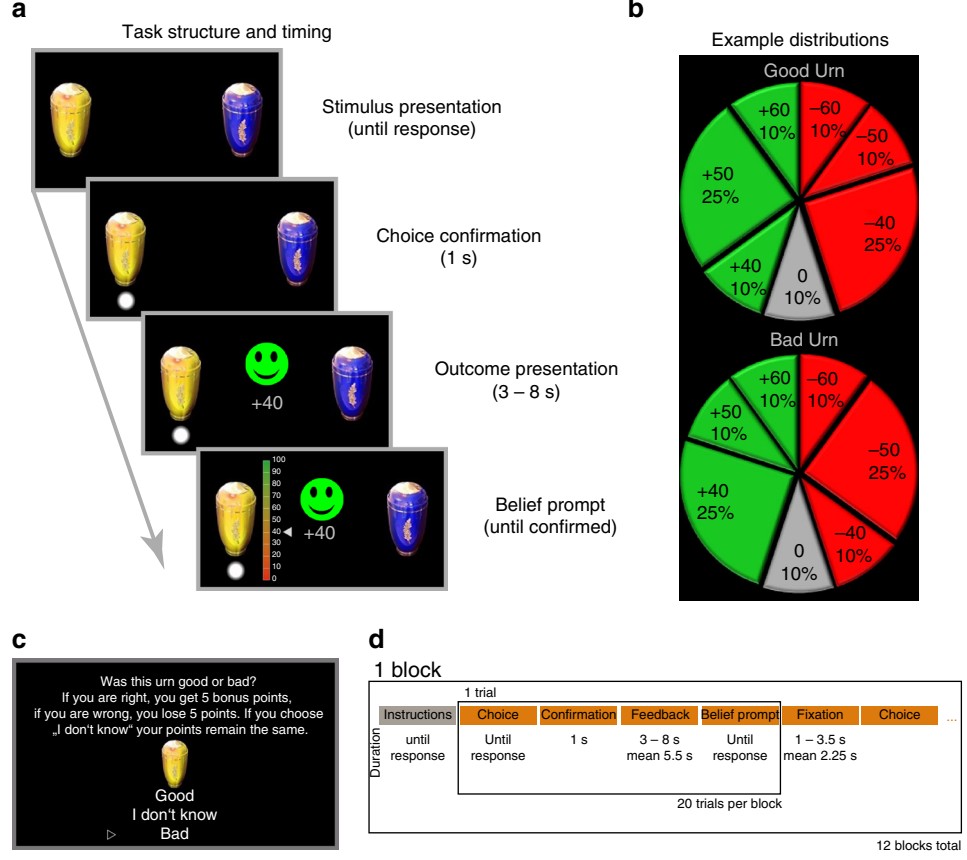

**Fig. 1** Task design. Schematic of an example trial and event timing of the task **a**. All participants were informed **b** about the exact probability of each possible event in two lotteries (represented by urns yellow and blue) in relation to their long-term valence by presentation of two pie charts. Good urns had a long-term expected value above and bad urns below zero. In this example **a**, a participant obtained a positive pay-out of 40 points, equivalent to 1.2€. However, this event is more likely to occur if the chosen urn indeed was associated with a long-term pay-out below zero (**b**, likelihood-ratio between urns 0.4). Therefore, a participant could infer that the likelihood of this urn to be good should be lowered by this experience, despite having just experienced winning 40 points. We prompted participants about their beliefs explicitly to the inferences performed (lowest display shows belief bar). Most events in the task (Table 1) carried long-term information as well as valenced short-term pay-out, which could either be congruent (short- and long-term valence align) or incongruent (short- and long-term valence mismatch). Between blocks of 20 trials (**d**), participants could earn bonus points if they correctly identified the true long-term pay-out of both urns (5 bonus points per correct answer). However, participants were also given the choice to avoid this gamble (**c**). On average, participants identified the true long-term valence of good urns correctly in 92 ± 2% and bad urns in 86 ± 5% and chose not to guess at all in 6 ± 3% of the cases. Note that the pie charts illustrating possible event distributions were constantly available to the participant in order to avoid high working memory loads which could interfere with learning and could render the task very difficult to solve

**Table 1 Event structure of the task**

| Short-term valence | Long-term valence | Informativity | Congruency | Event distribution | | Payout | | | |
|---|---|---|---|---|---|---|---|---|---|
| | | | | Good urn | Bad urn | Variant 1 | Variant 2 | Variant 3 | Variant 4 |
| Negative | Negative | Informative | Congruent | 10% | 25% | −50 | −50 | −30 | −30 |
| Negative | Positive | Informative | Incongruent | 25% | 10% | −40 | −40 | −20 | −20 |
| Negative | None | Non-informative | None | 10% | 10% | −30 | −60 | −10 | −40 |
| None | None | Non-informative | None | 10% | 10% | 0 | 0 | 0 | 0 |
| Positive | None | Non-informative | None | 10% | 10% | +30 | +60 | +10 | +40 |
| Positive | Negative | Informative | Incongruent | 10% | 25% | +40 | +40 | +20 | +20 |
| Positive | Positive | Informative | Congruent | 25% | 10% | +50 | +50 | +30 | +30 |

Payouts in the task varied over blocks, and can be categorised via the following criteria: informative events (column 3) had unequal probabilities to occur in good or bad lotteries (urns). Thus, they can be used to derive, or update (column 2), the conditional probability of a lottery to be good or bad in the long-term. This update was de-correlated (Fig. 2e) from the valence (positive or negative) of an event (column 1) and those events where the direction of long- and short-term valence align, are termed congruent (column 4), and incongruent where they mismatch. The long-term valence results from congruent events always having larger absolute payouts, which, however, in relation to the actual event is small (difference ± 10 points, maximal payout in the task 60 points per outcome). In addition, non-informative valence events did have the highest absolute payout in half of the blocks. The likelihood-ratio of informative events in good and bad lotteries (2.5 or 0.4) was constant throughout the experiment. In total 30% of possible outcomes did not carry information about long-term valence (rows three to five), and 10% additionally had no short-term outcome (payout of 0 cents, row 4), and these thus were non-informative. Note that in some blocks both urns could be good or bad, respectively (with the same pay-outs)

experience influences inference. Furthermore, we employed a Bayesian framework that allows us to compare human to normative, ideal inference[10].

To this end, we designed a novel binary learning task in which participants had to learn if the long-term valence of urn lotteries was good or bad in a block-wise fashion. All participants were given full information about the probability of events depending on whether a good or bad urn was chosen, and this information could be employed for inference. Both negative and positive outcomes could indicate that an urn was good, i.e., had a positive long-term expected value. We were thereby able to contrast the informational content of a trial from its current rewarding nature. For example, a specific rewarding outcome could be more frequent in the bad urn, leading to a positive reward prediction error (RPE) for the current outcome, and a (potentially Bayesian) inference that this urn is likely bad. To study belief formation directly, we prompted participants about their explicit belief estimates of whether or not an urn was good or bad in the long-term. This design allowed us to contrast actual inference from the experienced reward associated with an event.

Previous neuroimaging studies hinted at a common representation of both these short- and long-term learning mechanisms in the medial striatum[8]. These common effects contrast with findings on different mechanisms of motivated behaviour which have dissociated representations across different parts of the striatum[11–13]. We used functional magnetic resonance imaging (fMRI) to investigate how 24 young healthy participants assign the relative weight of reward experience and information about long-term net outcome to generate beliefs that guide behaviour. The novel two-urn task provided an ideal Bayesian solution to the inference problem and was easily accessible by the participants. This design allowed to assess how far individual participants deviated from optimal belief updating and to map short-term learning and inference about long-term outcomes onto computationally defined functions. The task fully orthogonalised both learning mechanisms and thus provided ideal preconditions to search for their possibly separable neural correlates. We found that human inference, measured via belief updates, was biased away from ideal Bayesian inference and towards short-term reward experience. On a neural level, we found a dissociation between reward representations and belief inference in the ventral to dorsal striatum, which we reproduced in an independent sample. Activity in the frontopolar cortex (FPC) and dorsal striatum counteracted reward-induced biases on inference. Furthermore, medial striatal representations of both learning mechanisms show gradients in opposite directions, and the individual degree of representational overlap of biasing short-

term rewards with model-based long-term inference in mediodorsal striatum corresponds to approximately optimal belief updating of that participant. In summary, we describe a novel task that can be employed to study the contribution of reward-biases on inference on neural and behavioural levels in healthy and possibly pathological states. The results we report elucidate that distinct regions in the striatum and FPC enable humans to draw reward-independent inferences.

## Results

**Task description and behaviour.** On every trial of the two-urn task (Fig. 1a), participants had to decide from which lottery (urn) they wanted to sample by drawing an event (a 'marble'). Every choice resulted in a pay-out of a specific number of points that directly translated into monetary incentives (10 points equalled 30€ cents), and which could be positive, negative, or zero. The immediate, short-term experience of pay-outs thus consisted in valenced monetary rewards, which ranged from 0 to ± 60 points. In addition, the exact value of the pay-out could be employed to infer whether an urn was good or bad to sample from repeatedly in the long-term. This inference should be based on the distribution of pay-outs conditional on the long-term valence of an urn. The event distributions of the possible urns were shown as pie charts (Fig. 1b) and continuously stayed on screen during the entire experiment to minimise working memory demands. In addition, before each block, participants were explicitly instructed about all possible events that could occur in the next block and how likely each event was in urns that were either good or bad. This part of the task was untimed so that participants could take the time they needed to understand the provided information. This ensured that they could infer (via the likelihood ratio of observing an event conditional on the long-term valence) if a specific pay-out indicated that the chosen urn was more likely to be good or bad. Thus, pay-outs simultaneously carried a short-term reward as well as information regarding whether the chosen urn was actually good or bad, constituting a potentially congruent or incongruent dual-signal (see Fig. 1 for an inference example). The same pay-outs could be observed when drawing from either urn, yet with different probabilities (Fig. 1b). This ensured that each event only provided a certain amount of information about the long-term valence of an urn and that participants had to integrate information via inference into a belief of how likely they thought an urn was to be good or bad. Notably, the short-term reward was irrelevant to solve the task in an ideal way, and ideal Bayesian inference would rely purely on the differences in the probability distributions.

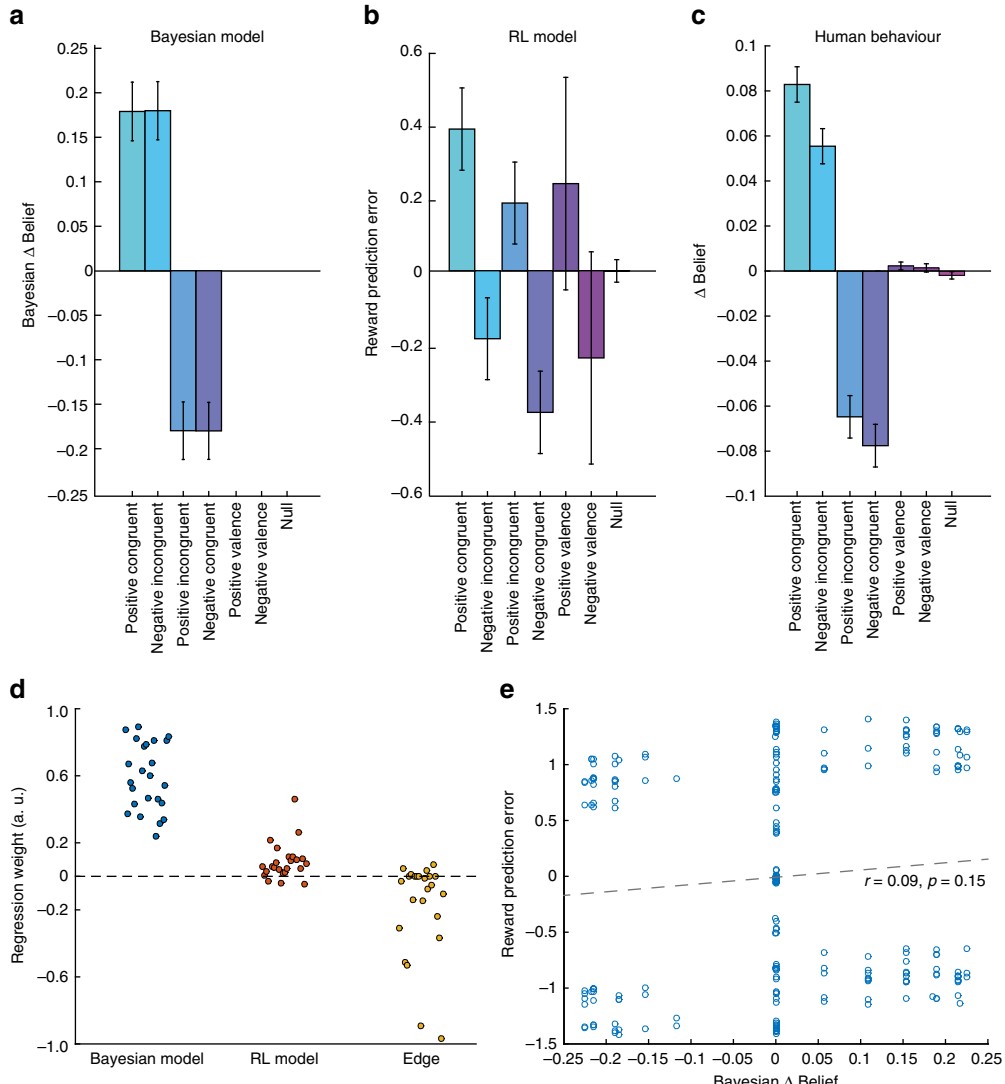

**Fig. 2** Modelled and observed behaviour in the task. The Bayesian learner **a** updates equally on congruent and incongruent informative events, and does not update on non-informative events (rightmost three entries). **b** Reward prediction errors (RPE) that update the model-free reinforcement-learner (RL) do not differentiate informative and non-informative events, and instead mainly reflect the valence of the outcome. **c** Human participants update their beliefs positively on positive congruent and negative incongruent events and vice versa for negative updating. They also do not update their belief estimates significantly on non-informative valence events, but instead show characteristics of biased updating by model-free learning: they update beliefs significantly less on incongruent pay-outs (see Supplementary Fig. 1 for a more in-depth analysis of the congruency effect). **d** Results of within-participant regression analyses comparing the influence of both learning models. All participants in the study were described by the update predicted by the Bayesian model, yet additionally a significant positive covariation between participants' belief estimate updates and the model-free RPE was seen. Factor edge controlled for the tendency of some participants to decrease updating at extreme values of possible beliefs (i.e., zero or one, $t_{23} = -3.0$, $p = 0.0063$). **e** Correlation of Bayesian belief update and normalised RPE across the task in one random example participant illustrating that both regressors were de-correlated. Predictions of both models are based on 5000 random trial sequences. Error-bars represent SD in plots **a** and **b**, and SE in **c**

After every trial, participants were prompted to enter their current beliefs about whether the urn was good or bad by setting a marker between 0 (bad) and 100 (good). Some events were equally likely in both types of urns, and therefore did not carry information. These are termed non-informative in contrast to pay-out events that are informative (Table 1). Note that non-informative events could still consist in winning or losing points, and thus provided reward experience but could not be used for inference. The long-term valence of an urn could be determined by calculating its expected value via multiplication of each event's probability and pay-out sum. This, however, was not something participants had to do because they were explicitly given information which distribution represented good or bad urns. To summarise, participants had to infer whether each of two urns

were good or bad via repeated sampling of events for which probability distributions were provided.

Participants could maximise their winnings in two ways. First, by preferentially choosing the good urns and avoiding bad urns within the blocks. Second, correct evaluation of the long-term valence of an urn was rewarded at the end of each block with bonus points if the true valence of an urn was correctly identified in an additional gamble (Fig. 1c). We introduced this bonus to encourage exploration of good as well as bad urns and keep the task, as well as analyses more balanced. The experiment consisted in 12 blocks of 20 trials (Fig. 1d). In 10 blocks, one urn was good and the other one bad. In one block, both urns were good, and in another block, both were bad, which was always the third block in the experiment to avoid that participants generalised beliefs of

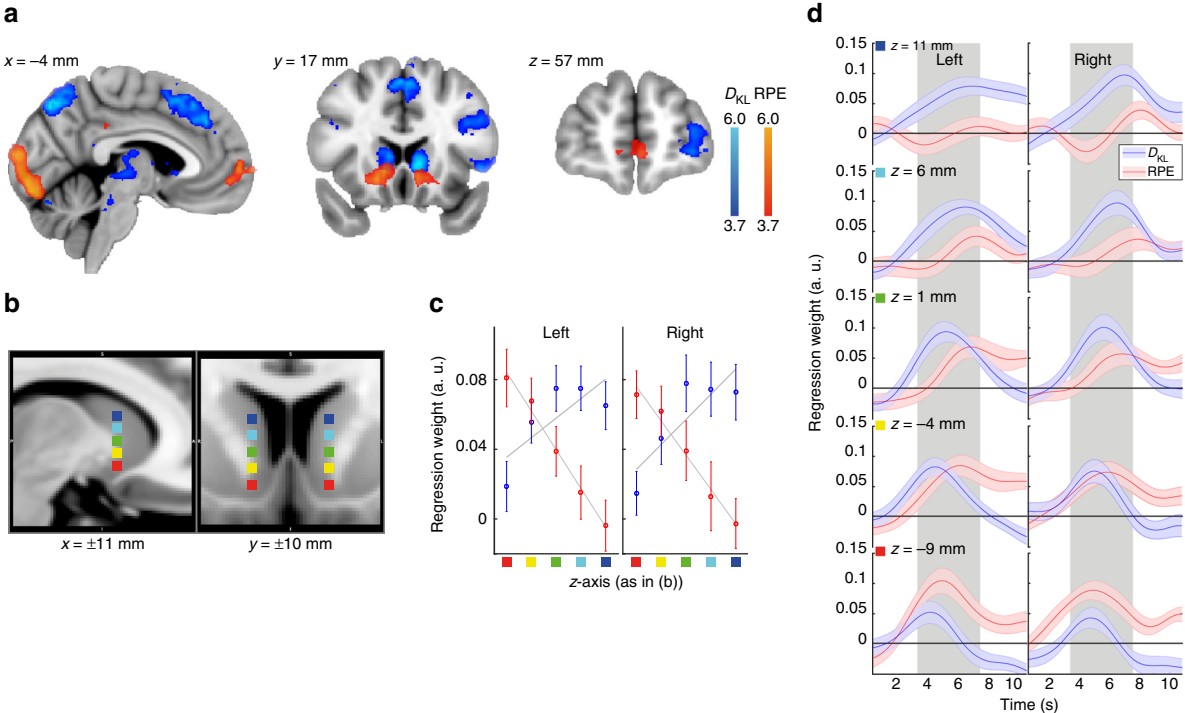

**Fig. 3** Effects of model-based inference and model-free experience. **a** Shows positive covariation with Bayesian model update ($D_{KL}$, blue) and reward prediction-error (RPE, red). $D_{KL}$ was associated with activity in the intraparietal sulcus (IPS; peak Montreal Neurological Institute coordinates −14, −70, 54 mm, peak z-score = 6.00), posterior mesial frontal cortex (pMFC; 0, 27, 50 mm, z-score = 5.40), left dorsolateral prefrontal cortex (dlPFC; −45, 25, 27 mm, z-score = 6.03), and left frontopolar cortex (FPC; −35, 60, 6 mm, z-score 5.07). In addition, the dorsal striatum (caudate nucleus) covaried with $D_{KL}$ (right 9, 17, 6 mm, z-score = 5.07; left −9, 16, 2 mm, z = 4.94), whereas the ventral striatum covaried with RPE (right 14, 9, −8 mm, z = 5.51; left −9, 13, −8 mm, z-score = 5.00), for which an additional effect was seen in the ventromedial prefrontal cortex (vmPFC −4, 61, 3 mm, z-score = 4.89) extending into the medial FPC. A complete list of activations can be found in Supplementary Table 1. Spatial gradient analysis comparing functional main effects against anatomical location along the bilateral z-axis from ventral to dorsal striatum (x = 11, y ± 10 mm) in 5 mm steps (indicated by the colour-marks in **b** that correspond to the z-axis in **c** and plotted in **d**). Within-participant multiple regression time-courses of beta weights of an analysis comparing BOLD activity within these marked voxels against $D_{KL}$ and RPE per trial (including regressors of no-interest). **c** displays mean regression-weights from 3 to 7 s (grey area) which are compared against the z-axis marked in **b** in an across-participants regression model. The time-course of this signal is plotted in **d**. In the left and right striatum, RPE representation gradually decreases along dorsal position (both p < 0.0005), while $D_{KL}$ representation increases (both p < 0.0125), resulting in significant contrasts within both hemispheres at most ventral and dorsal seed regions (all p values < 0.001). Effects in **a** are thresholded at p < 0.0001, colour bars indicate z-scores, error-bars in **c** and shades in **d** reflect SEM of individual participants' regression weights, lines in **c** reflect the OLS regression slopes of the averaged regression values for illustration purpose only. For an analysis of decision related neural correlates see Supplementary Fig. 6

one urn to the other one. The pay-out values and distributions changed from block to block, such that the same event could be informative or non-informative in different blocks. The expected value for each urn was constant at ± 30 points (equal to 0.9€) per block, meaning that if participants would only sample from the good urn, they would gain 1.5 points per trial. The exact order of events was predetermined with regard to the information conveyed (congruent, incongruent, non-informative), but the valence depended on which urn a participant chose (e.g., a congruent event would be positive if a good urn was chosen on that trial). Note that magnitude and relative pay-out between informative and non-informative events were varied across the task (Table 1), yet the likelihood ratio of informative events remained constant over blocks, which facilitated task performance.

Participants correctly identified the urns' long-term pay-out accurately and equally well for good and bad urns (good correct = 92 ± 2%, bad correct = 86 ± 5%, no significant difference, $t_{23}$ = 1.45 p = 0.16, n for these and all following tests unless specified otherwise = 24). This indicates that the participants understood the task well and could integrate outcomes into long-term beliefs by using the provided event distributions for good and bad urns. The latter result was likely due to the fact that we incentivised

participants to explore both urns and not purely exploit their current belief about which urn may have the higher expected value. We furthermore found that participants' beliefs mediated choices: the effect of experiencing positive and negative outcomes on beliefs fully explained choice behaviour (following inclusion of beliefs into a regression of learning duration on choices, more time on task had no effect on choices of correct urns, see Supplementary Fig. 2). This indicates that measuring beliefs is sufficient to explain choice behaviour and most biases in the two-urn task.

**Short-term outcomes bias inference-based learning.** Ideally, inferential long-term belief updating should be independent of the short-term valence of the pay-out and approach Bayesian optimal updating (Fig. 2a). To test whether participants achieved this ideal updating (or systematically deviated from it), we constructed two computational models that solved the task. The first one was a Bayes-optimal learner[14] that employed model-based information (Fig. 2a), constituting a normative solution to the task[10]. The second model relied on outcome information without knowledge about the underlying pay-out distributions, as in model-free reinforcement learning (RL, Fig. 2b)[15]. From the

Bayesian model, we extracted the signed trial-by-trial deviance between prior and posterior belief, which measures how much and in which direction beliefs should ideally be updated (Bayesian $\Delta$Belief), and which closely relates to the absolute change in posterior belief, the Kullback–Leibler Divergence ($D_{KL}$, see Methods for details). From the RL learner, we extracted the signed RPE and regressed both in one multiple linear regression onto the trial-by-trial belief update of the participants. This confirmed that all participants updated their beliefs well in accordance with predictions from the Bayesian model ($t$-test of within-participant regression weights $t_{23}=14.5$, $p<10^{-13}$, Fig. 2d), indicated by positive regression weights for each participant. However, RPE derived from the RL model, which represents the difference between obtained and expected pay-out based purely on previously experienced pay-outs, explained additional variance in belief updating ($t_{23}=3.86$, $p=0.0008$). This indicates that long-term beliefs are systematically biased by short-term reinforcement learning. This is also reflected in significantly lower belief updates when short- and long-term information are incongruent with each other ($t_{23}=4.06$, $p=0.00049$, Fig. 2c and Supplementary Fig. 1). We found a comparable effect when we used the actual outcome of the trial instead of the RPE ($t_{23}=3.76$, $p=0.001$), indicating that in this task RPE effects are mainly driven by the outcome term. Importantly, non-informative events that carried no long-term information, yet still provided valenced pay-out, did not lead to significant belief updating (both $t$-tests $p>0.19$)—ruling out the possibility that participants did not understand the task. Failure to understand task instructions would lead to updates always in accordance with the valence of the pay-out (Fig. 2c).

**Dissociable neural correlates**. To search for separable neural correlates of model-based and model-free learning, we included predictions of the models as trial-by-trial regressors locked to the onset of the pay-out into one general linear model (GLM) to explain the fMRI signal time course of every participant across the task. The predictors were $D_{KL}$[16], reflecting the overall degree of change in the Bayesian model, as well as the RPE, reflecting signed model-free outcome evaluation. Inclusion into one model was unproblematic, because both predictors shared only 1.2% variance (average $r$ across all participants = 0.11, Fig. 2e).

Our results revealed evidence for separate neural representations of both learning mechanisms as well as for partial overlap in distinct cortical and subcortical regions. We found that $D_{KL}$ was associated with activity in the intraparietal sulcus (IPS; peak $z$-score = 6.00, Fig. 3a, please see Supplementary Table 1 for additional details for all effects), posterior mesial frontal cortex (pMFC; $z$-score = 5.40), left dorsolateral prefrontal cortex (dlPFC; $z$-score = 6.03), and left FPC ($z$-score = 5.07), as well as dorsal striatum (caudate nucleus) (right $z$-score = 5.07; left $z$-score = 4.94). The ventral striatum expectedly[17] covaried with RPE (right $z$ = 5.51; left $z$ = 5.00), for which an additional effect was seen in the ventromedial prefrontal cortex (vmPFC; $z$-score = 4.89) extending into the medial FPC.

Contrasts revealed that distinct cortical activity covarying with model-free learning signals was seen in the vmPFC ($z$-score = 5.3, Supplementary Fig. 3b) and ventral striatum (left $z$-score = 4.07; right $z$-score = 3.21). All $D_{KL}$ effects were significant above RPE effects (contrast at peak voxel in IPS $z$-score = 5.04, dlPFC $z$-score = 5.24, pMFC $z$-score = 4.41). We furthermore confirmed that all $D_{KL}$ main effects were seen over-and-above the possible influence of surprise (formalised as Shannon Information, $t$-test of regression weights at peak voxels all p < 0.003, Supplementary Fig. 3c–f). Shannon-Information is a possible confound in analyses investigating Bayesian model update[16], because in most

cases, unexpected events cause larger changes in beliefs. This confirms dissociable neural correlates for model-free and inferential learning. However, both processes were not completely distinct. A cluster conjunction analysis of both learning-models' predictions showed overlapping activity in the medial striatum at a minimum $z$-score threshold > 3.1, as well as two cortical regions (Supplementary Fig. 3a), including the lateral FPC and posterior cingulate cortex (PCC).

**Striatum gradually reflects both learning signals**. We then investigated whether the observed difference in striatal correlates of $D_{KL}$ and RPE could be described by assuming a gradual in- and decrease of one type of learning over the other, or reflected a strict anatomical dissociation. Therefore, we conducted a spatial gradient analysis and regressed the position along the ventro-dorsal axis of the striatum (Fig. 3b) against the regression weights reflecting covariation with either $D_{KL}$ or RPE. We found that within both hemispheres the in- and decreasing expression of Bayesian update and model-free learning, respectively, was well described by assuming gradual change (Fig. 3d), which was confirmed by significant regression slopes with reversed signs for both factors along the MNI $z$-axis (Fig. 3c, $t$-test of slopes in both hemisphere for $D_{KL}$ $t_{23}<-2.7$, $p<0.015$; RPE $t_{23}>4.1$, $p<0.0005$). Although it is difficult to fully rule out artefacts as the cause of gradual effects in fMRI, this finding is compatible with previous research suggesting that information is integrated from ventral to dorsal striatum via spiralling reciprocal ascending midbrain projections[18].

Next, we tested if the effect for model-free signals obeyed to sufficient criteria[19] of a true prediction error signal. These criteria demand that the actual pay-out correlates positively and the expectancy of the pay-out (the expected value) negatively with the signal. Yet, whether this expected value reflected in the ventral striatum—which forms the basis for RPE calculations—is learned purely via model-free mechanisms or incorporates more complex learning strategies, is currently unclear[8]. Therefore, we included the actual pay-out obtained and its expectancy (defined either as learned via the RL model, or as a participant's actual prior belief) into one GLM focused on the ventral striatum. This analysis revealed that, aside from the pay-out, ventral striatum negatively covaried (left $z$-score = $-2.21$, right $z$-score = $-3.09$) with the participant's current belief, rather than the RL model's expected value (both $p>0.1$, Supplementary Fig. 3g, h). This suggests that model-free and inference-based learning are intertwined processes which could be related to information transfer to the striatum, specifically when learned outcomes need to be unified with instructions[20]. Thus, regions assumed to reflect mainly model-free, short-term outcomes can be informed about the current expected value by model-based learning and compute an outcome prediction-error in relation to model-based values[8] or beliefs. In addition, we found that the dorsal striatum represented current beliefs at outcome relatively weakly (right caudate $z$-score $-2.67$, left $z$-score = $-0.43$, Supplementary Fig. 3c–f), suggesting that update terms and expectancy are at least partially processed by different brain regions.

**Correlates of more ideal updating**. To further characterise the trial-by-trial neuronal activity associated with more or less optimal updating, we included a regressor coding the distance between a participant's actual update (prompted later) and the prediction from the Bayesian model. Thus, *Bayesianness* was quantified as the trial-by-trial similarity of reported updating to ideal updating such that 1 equals exact Bayesian updating and 0 the opposite. By adding *Bayesianness$_t$* to the fMRI GLM, we seek regions in which activity predicts more ideal belief updating over-and-above the

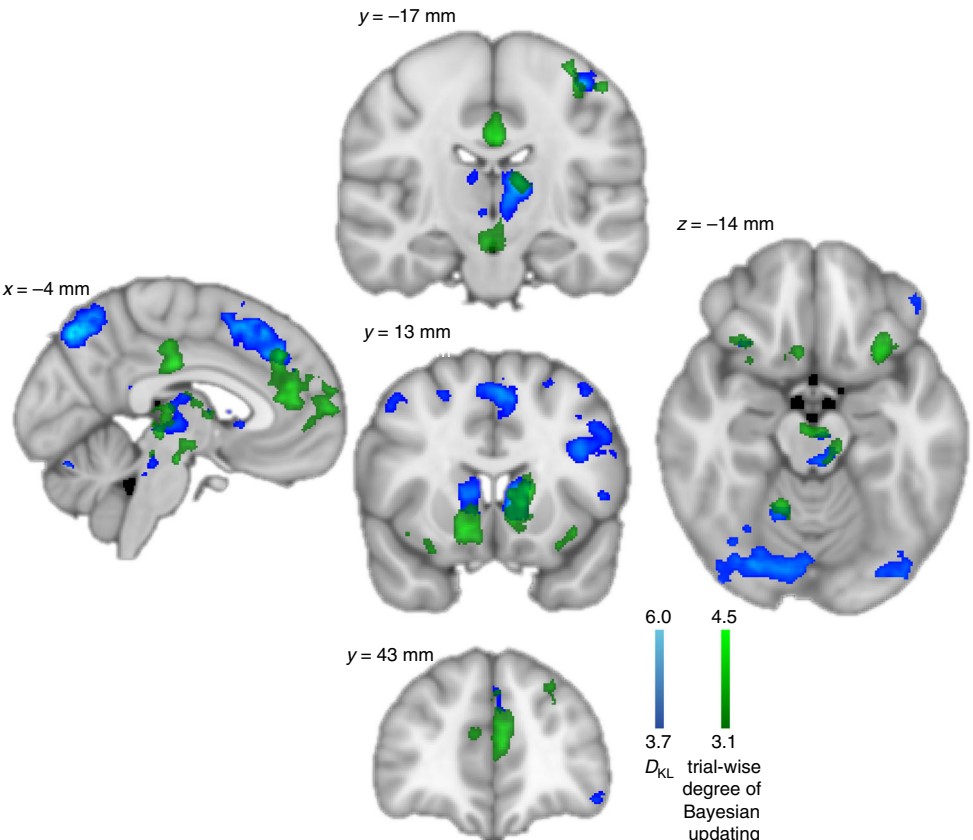

**Fig. 4** Trial-by-trial analysis of the degree of ideal Bayesian updating. Green = significant positive effects for regressor *Bayesianness*, indicating regions where BOLD signals increased whenever belief updates were closer to Bayesian ideal updating. Notably, activity in ventral (right 14, 13, −9 mm, $z$-score = 4.27) and dorsal striatum (left 14, 13, −9 mm, $z$-score = 4.27), FPC (−4, 62, 9 mm, $z$-score = 4.05), anterior midcingulate cortex (aMCC, −9, 36, 27 mm, $z$-score = 4.71), posterior cingulate cortex (PCC, −1,−19, 32 mm, $z$-score = 4.65) and midbrain overlapping with the substantia nigra (SN) and ventral tegmental area (VTA, −2, −12, −12 mm, $z$-score = 3.80), was found to be higher whenever a participant's update was closer to optimal. See Supplementary Table 1 for details about significant cluster activations. All these effects at least partly overlapped with main effects of $D_{KL}$ (reproduced as in Fig. 3, depicted in blue) despite whole brain FWER correction. Colour bars indicate $z$-scores, plots are cluster extent corrected at $p < 0.05$

bias of model-free learning (RPE), and in which activity varies across the task that is unexplained by the Bayesian model itself[21].

The degree of optimality of an update on a given trial positively covaried with activity in aMCC ($z$-score = 4.71), bilateral striatum ($z$-score > 4.34), and left lateral FPC ($z$-score = 4.29), all overlapping with the main effect of $D_{KL}$ (Fig. 4, see Supplementary Table 1 for all results). This indicates that when these regions are activated more strongly on a given trial, updating is closer to the optimum, possibly because the model-free bias is reduced. A notable additional finding are partly overlapping effects of update optimality (peak coordinates −2, −19, −18 mm, $z$-score 4.05) and $D_{KL}$ (peak coordinates 0, −26, −22 mm, $z$-score 4.28) in brain stem nuclei, indicating that activity here increased as optimal updating increased. Although we lack the resolution to differentiate between individual nuclei, we note overlap possibly with the ventral tegmental area (VTA), substantia nigra (SN), or raphe nuclei. This finding is physiologically plausible given the tight reciprocal connections between these midbrain regions and the ventral and dorsal striatum[18].

We further tested for an association with incongruence between model-free reward experience and long-term inference. First, we found that IPS ($z$-score 4.7), dlPFC ($z$-score 4.07) and pMFC ($z$-score 3.54) displayed increased activity when reward experience and long-term inference were incongruent with each other (Supplementary Fig. 5), consistent with a role of these regions in implementing cognitive control and suppressing model-free biases[22–24].

**Neural predictors of learning performance**. We then asked whether inter-individual differences in neuronal representations of model-free and model-based learning relate to the ability to overcome model-free learning biases. Importantly, just as in the introductory examples relating to choosing fast-food or exercise, we observed considerable variance in the ability to adhere to ideal Bayesian updating across participants (Fig. 2d), as well as in the bias induced by model-free learning. We therefore regressed individual participants' regression weights of Bayesian and RL models on belief updating obtained from the behavioural regression analysis (Fig. 2d) onto the contrast (Supplementary Fig. 3b) of parameter coefficient estimates for $D_{KL}$ and RPE in a whole brain analysis. Resulting parameter coefficients were then again contrasted; therefore this contrast of contrasts tests the hypothesis of any relationship between neural data and efficacy of belief formation and influence of model-free biases. Surprisingly, we found that the lower the difference between coding $D_{KL}$ and RPE was in bilateral dorsal striatum ($z$-scores > 4.41), as well as FPC ($z$-score = 4.59, Fig. 5a), the more was a participant's belief update explained by Bayesian compared to model-free learning, i.e., the more optimal and the less biased was a participant. We conducted separate cluster-based conjunction analyses (at $z$-score threshold > 3.1) to test if this across-participants effect overlapped with regions revealed in the within-participants analyses as coding: either $D_{KL}$, RPE, or both. There was some overlap within the medial striatum indicating conjunct effects for RPE and behavioural correlation across participants, yet the overlap

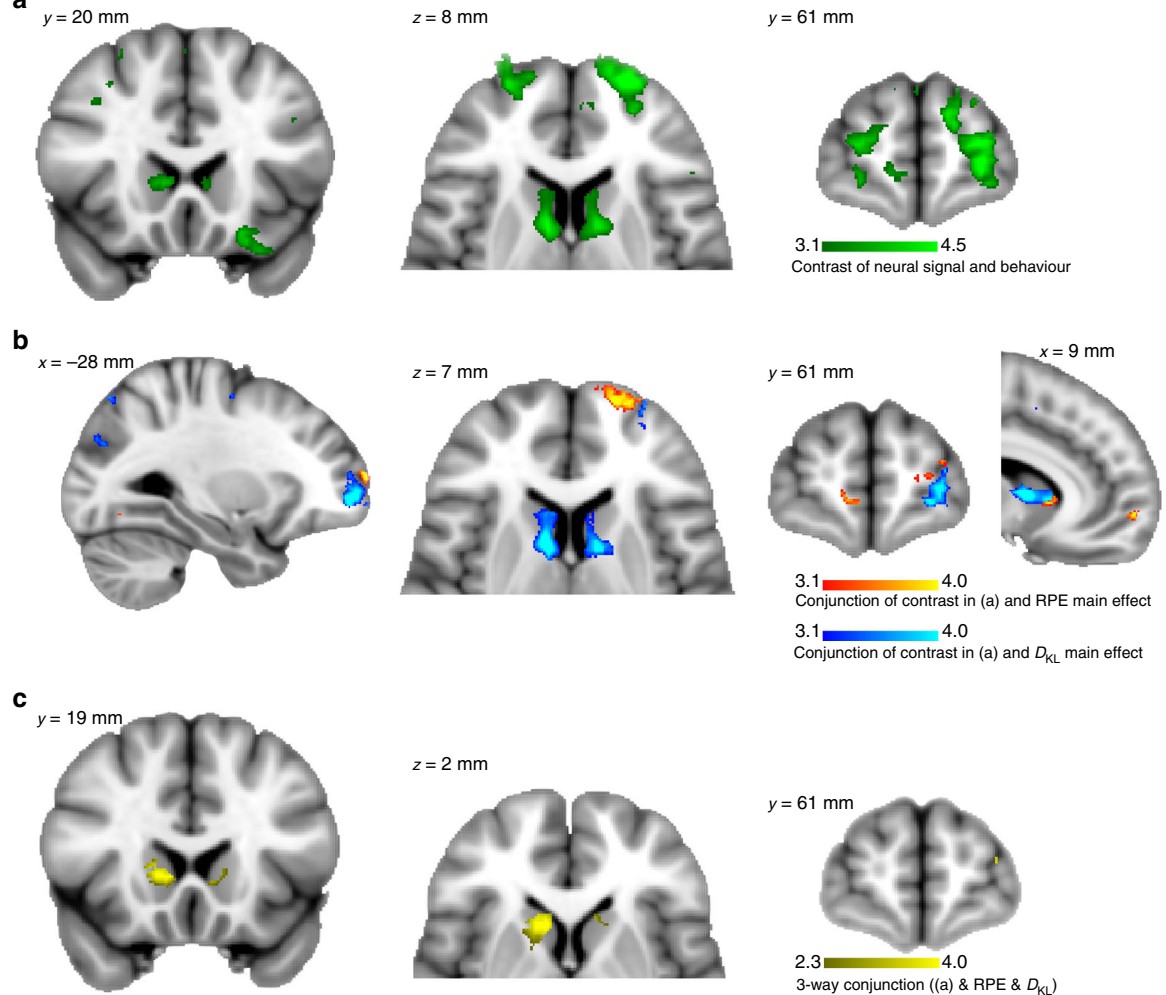

**Fig. 5** Striatal and FPC activity predicts Bayesian updating performance. **a** Displays the effect of contrasting behavioural and neural effects across participants forming a contrast of contrasts. The lower the difference between neural short-term (RPE) and long-term ($D_{KL}$) representations in bilateral caudate (right 9, 2, 10 mm, z-score = 4.43; left −11, 3, 13 mm, z-score = 4.41) and FPC (right 17, 66, 5 mm, z-score = 4.34; left −24, 64, 9 mm, z-score = 4.59), the more participants updated their beliefs in accordance with predictions from the Bayesian, rather than the RL model. Conjunction analyses **b** with the main effects of RPE (red) and $D_{KL}$ (blue) showed overlap mainly within regions that were primarily activated by Bayesian model update, although medial striatum and FPC activity additionally overlapped with the main effect of RPE. This was confirmed by a three-way conjunction analysis which revealed that indeed the medial striatum, as well as a small region in FPC was significantly activated by model-free and inference-based learning and reflective of the degree of influence both models exerted over participants' belief updates when compared on group level (**c**, yellow). Colour bars indicate z-scores, plots are cluster corrected at $p < 0.05$

was clearly pronounced in the dorsal striatum with the main effect of $D_{KL}$ (Fig. 5b). Last, the three-way conjunction (at a minimum z-score of 2.3) revealed that the medial striatum was activated by both main effects, as well as displaying significant across participants' covariation with the relative degree of Bayesian belief updating (Fig. 5c).

To test whether this contrast of contrast effect was driven by increased RPE, or decreased $D_{KL}$ representations, as well as whether it was mainly related to the influence of Bayesian or RL model predictions, we conducted separate follow-up analyses across participants on the respective main effects. We found that increased expression of model-free learning signals in dorsal striatum (z-scores > 4.05) and FPC (z-score > 3.98) drove the behavioural contrast (Supplementary Fig. 4), which may be activity indicating suppression of reward-related signals. Therefore, it was not the relative degree of expression of Bayesian learning signals that explained inter-individual variance, but the representation of model-free signals within regions coding

inferential model update. Furthermore, when dissociating the effects in the behavioural data, we found that behavioural correlations with striatal activity were mainly related to a reduction of the influence of RL learning on belief updating (z-scores > 3.80, Supplementary Fig. 4c). Within the FPC, we found dissociable effects such that stronger representation of RPE in medial FPC was related to a reduction of model-free influences (z-score = 4.17), and in more lateral FPC related to increasingly Bayes-optimal belief updating (z-score = 4.18, Supplementary Fig. 4c).

We furthermore examined whether inclusion of RPE coding into more dorsal striatal regions was associated with a reduction in biased belief updating, thereby excluding the possibility that this effect was introduced by anatomical variance across participants. To do this, we regressed individual regression-weights of model-free and model-based learning on belief update (Fig. 2d) onto the striatal gradients from ventral to dorsal regions for RPE and $D_{KL}$ representations (Fig. 3c) averaged over left and

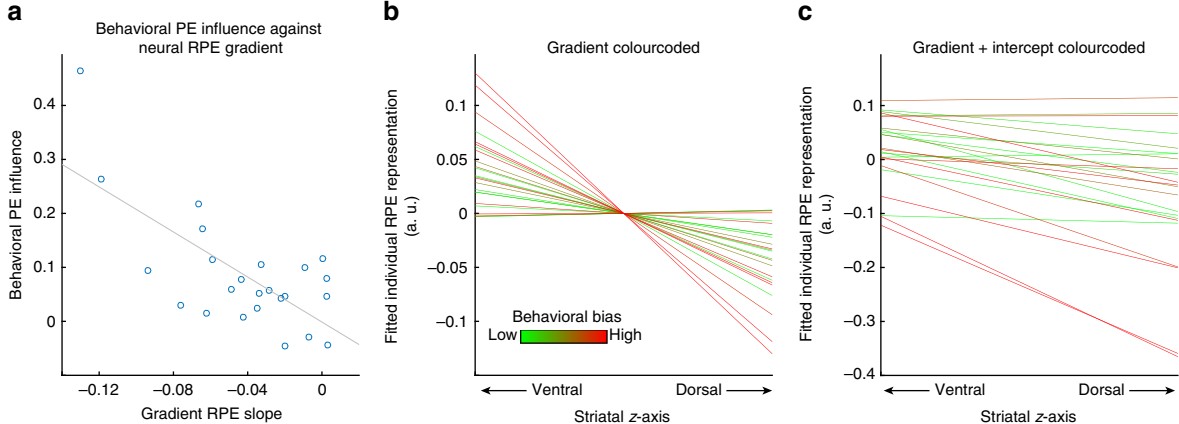

**Fig. 6** Gradual RPE integration into the dorsal striatum reduces model-free biases. **a** Scatter plot of the overall correlation ($r = -0.70$, $p = 0.00012$) between the individual slopes of the striatal gradients of RPE representation (as in Fig. 3c; averaged over both hemispheres) and the individual behavioural influence of RPEs on belief updating (regression weights displayed in Fig. 2d middle). **b** Individual striatal gradient slopes (averaged over both hemispheres), colour-coded according to the individual rank order of behavioural bias by RPE (short-term outcomes) on belief updating (red = strong short-term bias, green = low short-term bias). **c** Same as **b** with the intercept of the individual regression added to the plotted regression lines. In sum, weaker representations of the RPEs in dorsal striatum were associated with stronger biases of a participant's behaviour by model-free processing of short-term outcomes

right hemispheres. We found a significant effect of the behavioural influence of model-free predictions and the steepness of the RPE gradient (Fig. 6): the more RPE signals were focused ventrally and spared the dorsal striatum (steeper negative slope), the more model-free learning biased belief update (robust regression $t_{20} = -3.94$, $p = 0.0008$). There were no significant effects (robust regression all $p > 0.12$) for model-based updating on RPE or $D_{KL}$ gradient slopes, nor for model-free updating on the slope of $D_{KL}$ gradients. These findings suggest that individual differences in the degree of the efficiency of long-term belief formation are not explained by variance in expressing model-free learning signals in the ventral striatum, nor the degree of representing Bayesian updating in dorsal striatum per se. Rather, it seems that representation of model-free learning within dorsal striatum is essential for unbiased formation of beliefs about future outcomes, which functionally may represent repression or counteracting of model-free signals.

**Replication study**. Finally, we performed a replication study in an independent sample of 18 participants. We altered the task in two ways: participants were only prompted about their beliefs on every third trial, and the belief markers always returned to the indifference point. Therefore, participants could predict when they would be prompted about their beliefs and they had to memorise the previous position of the belief bar. This ensured that participants formed internal belief representations and tested if simply anticipating that the position of the belief-bar would be updated could explain the neural effects.

Behaviourally, participants successfully identified the urns' long-term valence between blocks to obtain the bonus points (Fig. 7a). For good urns, they were correct $10.9 \pm 0.3$, wrong $0.6 \pm 0.2$ and for bad urns they were correct $11.1 \pm 0.2$ and wrong $0.3 \pm 0.14$ times. None of these numbers were significantly different from the original study (independent sample $t$-test, all $t_{40} < 1$, all p for comparisons $> 0.2$). We replicated the striatal dissociation found in the original study (Fig. 7b). We also replicated all major main effects seen for $D_{KL}$ and these overlapped with the $D_{KL}$ effects in the original study. We found significant effects in the left ($z = 3.72$) and right dorsal striatum ($z$-score = 4.06), left IPS ($z$-score = 4.07), left dlPFC ($z$-score = 4.9), pMFC ($z$-score = 4.6), and left FPC ($z$-score = 3.61). All these effects remained

significant when we analysed only trials without a following belief prompt (all $z$-scores $> 3.1$, Fig. 7c).

The effects for RPE were somewhat reduced, possibly because of the worse fit possibility of the RL model because prompts were present only every third trial. At a lower threshold, we still found overlapping significant effects in the left (Fig. 7d, $z$-score = 2.50) and right ventral striatum ($z$-score = 2.77). Furthermore, we found significant effects overlapping with the original cluster spanning vmPFC ($z$-score = 2.26) and FPC ($z$-score = 2.67). The only significant interaction effect between belief prompting and $D_{KL}$ or RPE was seen in the left IPS ($z$-score = 3.3), indicating that $D_{KL}$ was more strongly reflected here when participants knew that they would be prompted to enter their current belief estimate afterwards.

This supports the view that the observed neural correlates of belief updating relate to learning, although learning and the formation of a decision about long-term valences are closely intertwined processes and it cannot be excluded that part of the neural signals relate closer to decision making compared to learning[25]. However, long-term inference and model-free reward processing clearly demonstrated separable neural correlates, suggestive of dissociable neural processes underlying both.

## Discussion

As in the two-urn task, many real-life events carry multiple meanings: in addition to a reward or punishment experience they often convey information enabling us to update our internal beliefs about the world and thus enable us to infer whether a situation or course of action is generally favourable. Theoretically, ideal inference from an event should be independent of the reward experience it conveys. Our study suggests that the human brain processes reward experience and inference about long-term outcomes in distinct cortical and subcortical brain regions. The network related to inference was dissociable from model-free reward processing, and consisted of the dorsal striatum, frontal cortex (dlPFC, pMFC, FPC), and IPS; reward itself was processed in ventral striatum and frontal cortex. These distinct networks overlapped in the dorsal striatum and FPC, two regions that enable adaptive behaviour[13,26]. By studying belief formation via inference, we uncovered a cortical network that is similar to networks previously identified when a task model had to be

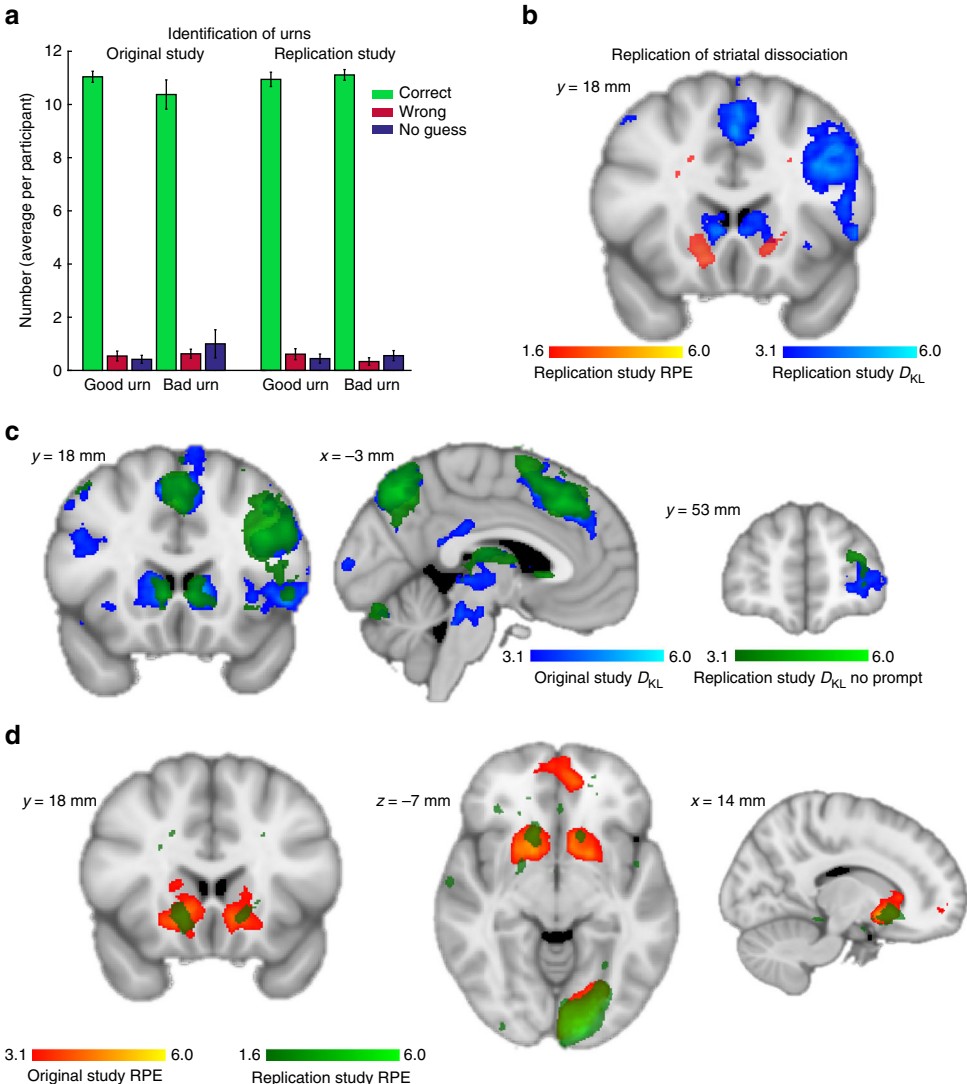

**Fig. 7** Results of the replication study. We replicated the main results of the study in a second sample of 18 participants. In this version of the task, participants knew beforehand that they would be prompted about their beliefs only every third trial. In addition, the belief marker was always reset to the indifference point, forcing them to always memorise their beliefs. Participants validly identified if urns were good or bad in the long-term **a**. We replicated the striatal dissociation (**b**, blue = $D_{KL}$, red = RPE), as well as all $D_{KL}$ main effects (**c**, original = blue, replication = green) even on trials where participants knew they would not be prompted about their beliefs afterwards. This minimises the possibility that $D_{KL}$ effects are related to downstream decision making about how to move the prompt cursor rather than to belief updating. RPE effects were weaker, but still present at lower thresholds, possibly because the RL model could not be fit to participants without prompting their beliefs. Colour bars represent z-scores, replication effects are displayed at corrected $p < 0.001$ for $D_{KL}$, and uncorrected $p < 0.05$ for RPE effects, original effects as in Fig. 1. See Supplementary Note 1 and Supplementary Methods for additional details about the replication study

explored[7], or model-updating was dissociated from simple surprise[16].

Within the striatum, we found that model-free and inference-based update parameters were represented in two independent spatial gradients of reversed directions. Model-free reward signals were strongest in the ventral striatum and decreased in dorsal direction, whereas model-based $D_{KL}$ was represented most in the dorsal caudate and least in the ventral striatum. Consistent with the idea of information transfer to the dopaminergic system[27], we found that the ventral striatum integrates beliefs and pay-out. This finding suggests that the ventral striatum does not reflect a learning mechanism that is strictly model-free, i.e., based purely on reward experience, but incorporates beliefs obtained by inference, compatible with previous studies[8]. Relatedly, we found that the dorsal striatum only weakly represented current beliefs at outcome while reflecting updating,

indicating that different brain regions may process values and update terms[28].

As a side note, we found that activity in dopaminergic midbrain regions, likely including VTA and SN, covaried with belief-updating as well as the degree of how close to Bayes-optimal an update was. It has previously been found that activity in dopaminergic midbrain regions increases with task demands[29,30]. Given that inference requires cognitive effort, one may speculate that increased midbrain activity could support ventral-to-dorsal information transfer in the striatum. This is anatomically supported by reciprocal connectivity between the striatum and midbrain regions, in which it is hypothesised that information is integrated into planning via spiralling projections from ventral to dorsal striatum and VTA to SN, respectively[18]. This speculative, new hypothesis awaits to be tested in pharmacological challenge studies.

Conceptually, the differentiation between reward experience and inference is related to studies investigating model-free and model-based learning[8], because inference has to rely on a model. However, the term model-based learning has been used broadly for many situations in which any kind of internal model was required. Most likely our investigation of abstract inference via probability distributions is qualitatively distinct from, for example, stimulus generalisation to obtain prospective rewards[9]. Therefore, we note that neural correlates of model-based learning likely depend on the actual model employed. Different models may explain why previous studies using the two-step task[8] (in which participants have to employ a model of state transitions) found common neural correlates of model-free and model-based learning only in the striatum. Although the two-step task allows to study inter-individual variance in model-based learning capacities[31] as well, neural activity induced by the two-step task has not shown the dissociation between ventral and dorsal striatum we report here. Another unique aspect of the two-urn task is that the model was always accessible in the form of the probability distributions, and learning consisted in applying the models to enable inference, whereas in most other tasks, the model itself has to be learned. Using Bayesian inference as a reference framework allowed us to compare behaviour and neural activity to mathematically ideal predictions, constituting a normative framework[10]. In contrast, ideal strategy employed in other tasks may, under some circumstances, be to not apply a model at all[32]. In sum, these specific properties of the two-urn task can explain why it allowed us to successfully dissociate the roles of the ventral and dorsal striatum in inference- and reward-based learning. To further investigate if this effect reflected general inference, or was additionally related to single-trial inferential processes, we compared participants' inference with ideal Bayesian inference. In doing so, we found brain regions that, on a trial-by-trial basis, displayed associations with the degree to which belief updating adhered to predictions of the Bayesian model over-and-above the effect of overall model-updating ($D_{KL}$). This revealed, that activity in the medio-dorsal and ventral striatum was associated with inferences closer to the Bayes optimum. In addition, increased activity in aMCC and FPC, both known to be involved in cognitive control, suggest that trial-wise increases in cognitive control overcome inference biases[33]. This effect was accompanied by increased activity in the PCC—speculatively associated with attentional focus on future decisions[34] and change detection[35]. dlPFC neurons have been shown to encode rewards that had not been directly experienced before, but had to be inferred by monkeys[36]; taken together, these results suggest that these regions contribute different facets to facilitate inference. These within-participants effects line up with between-participants effects. We found that inter-individual variance in the ability to infer ideal long-term beliefs about the world relates to the representation of model-free learning within regions associated with inference, specifically the dorsal striatum and FPC. In other words, the more RPE representation extends into the dorsal caudate and the more it is reflected in the FPC, the more optimal are an individual's belief updates. This may seem counterintuitive at first glance: common representation of model-free and model-based update parameters in the medio-dorsal caudate decreases the bias immediate reward exerts on belief updates. Similarly, more overlapping representations of RPE and $D_{KL}$ in (lateral) FPC was associated with more optimal belief updates, consistent with FPC's role in arbitrating between decision making strategies in other domains, such as exploration vs. exploitation[26,37,38]. However, this pattern of activity increases paralleled by inferential updating and reward signals related to the outcome, is well compatible with suppression of model-free outcome signals. Counteracting outcomes thus seems to enable

some participants to overcome belief-formation biases. These results fit nicely to a recent study in which participants could employ information conveyed either via real or hypothetical outcomes, which both conveyed the same information[39]. Participants' inferential learning was biased by real outcomes, yet activity in lateral FPC and pMFC counteracted this bias. The exact mechanism explaining how competition of short-term and long-term information could be solved in overlapping or neighbouring neuronal ensembles in the medio-dorsal caudate and the FPC remains to be elucidated in future studies. These should integrate biophysical models of learning and decision making with knowledge about cortico-striatal circuits[40], as well as information transfer between different striatal sub-compartments, possibly via spiralling connections through the dopaminergic midbrain[18]. Taken together, these studies showing dynamic, within-participants variance in belief-update and across-participants covariation with reward-induced biases on inference, suggest that cortical control mechanisms in frontal regions can overcome and suppress model-free learning biases, converging in processes reflected in striatal activity.

A functional dissociation between ventral and dorsal striatum has been associated with the formation of habits, i.e., behaviour resilient to outcome devaluation[41]. In fact, the dorsal striatum is supposed to control habitual responses. Interpreted in this framework, the activation in dorsal striatum we see may reflect formation of a habitual response that could, over time, replace active inference. It would be informative to test this hypothesis in future studies by exposing participants to the same event distributions over longer time periods. This would allow participants to directly associated events with belief changes, possibly bypassing active inference.

The current task shares certain features with the Iowa Gambling Task (IGT) which has been used widely to study the ability to integrate short-term outcomes into long-term predictions[42]. However, the IGT cannot deconstruct the intrinsic dual aspects of events which contain information, as well as rewards, because participants do not have an explicit model of underlying outcome distributions. Although some participants might construct a (subliminal) model in the IGT, it cannot be studied, when or if participants employ this model, and how it translates into behaviour. Imaging studies of the IGT indeed found broad involvement of the ventral and dorsal striatum and frontal cortex[43], yet specific contributions of these regions to learning cannot be described using the IGT. Furthermore, we found that most of the inter-individual variance maps onto neural correlates of short-term outcome processing, which renders an explanation based on individual processing capacity or task understanding unlikely. Certainly, future studies should address which individual factors map onto model-based and model-free learning in the two-urn task, as well as how neurotransmitter levels may modulate both forms of learning individually[44]. In sum, the two-urn task, which fully disentangles model-based and model-free belief updating, is an easy-to-perform task and a promising tool for clinical research on mental disorders. Future studies should additionally disentangle which brain areas convey learned value signals directly and which implement decisions based on values.

Finally, our results are highly relevant for economic decision theory. The economic consensus is that Bayes's rule is the only sensible way to update beliefs. Yet, even when full knowledge of the informational context has been made available to participants, we observed behavioural and neural biases. Thus, even without uncertainty which in theoretical decision-sciences motivated the introduction of post-Bayesian models[45], these types of biases should be accounted for in decision theory. Why humans are biased by short-term outcomes at all even when they should optimally be ignored remains an open question. Relying on

model-free outcomes is computationally inexpensive, such that biases of model-based behaviour might be related to limited processing capacities, or biases may have proven advantageous at some stages of human evolution. Our paradigm provides a new means to study whether belief formation is biased by recent events, allowing for the potential assessment of pathological deviations, an extension which could benefit the study of mental disorders such as attention-deficit hyperactivity disorder. On a neural level, we show that reward processing and inferential learning can be dissociated and that dorsal striatum and FPC enable reward-independent long-term inference.

## Methods

**Participants**. Out of the 26 young, healthy human participants that partook in the experiment, 24 participants were included into the final analysis. For one participant, the fMRI recording failed due to a broken head-coil, and one participant did not understand the task instructions (continuously selected the urn presented on the left side of the screen, leading to completely random choices). The final, included sample comprised 10 female participants and the mean age was $26.1 \pm 4.3$ (SD) years (range: 18–34). Two participants were left-handed and all participants gave written informed consent to participate in the study. Participants were informed about possible risks of the measurements prior to participation, and all procedures were carried out in accordance with the declaration of Helsinki. The study protocol was approved by the ethics committee of the medical faculty of the Otto-von-Guericke-University, Magdeburg (Germany). Details about the replication study are presented in the Supplementary Methods.

**Task description**. The goal of the task was to infer if a lottery, represented by an urn, had a long-term expected reward above (good urn) or below zero (bad urn) and use this information in order to maximise received pay-outs. Participants were instructed to deliberately choose between two urns to draw an event from at the beginning of each trial, and urn positions (left, right) were randomised. Following a 1 s delay period, the pay-out was revealed to the participant indicating either a monetary win or loss in points (range: 0–60 per trial) for 3–8 s (drawn from a uniform distribution with 1 s steps). At the end of the experiment, gained points were converted to € cents by multiplication with 3 and paid out to the participant. After the outcome was displayed, a belief prompt was shown in which participants had to enter their current belief about the previously chosen urn to be good (100) or bad (0) with a step-size of 10 by moving a marker up and down on the side of the belief-bar (Fig. 1a). The urns were selected by pressing a button with the left and right index finger, which also moved the belief marker up and down. The final position of the belief bar was confirmed with the left or right middle finger and trials were separated by display of a fixation cross for 1–3.5 s (drawn from a uniform distribution with 0.5 s steps). Participants were instructed that the prior likelihood of an urn to be good or bad was 50% and the marker remained at the previously confirmed position at the next trial when the same urn was chosen again (note the difference compared to the replication study).

To determine the long-term valence of each urn, participants were instructed to employ information about the conditional probability of each pay-out given that an urn was either good or bad (Table 1). Informative pay-outs conveyed signals on two dimensions: long- and short-term valence, which could align (congruent event), or mismatch (incongruent event). The short-term valence simply reflects the actual pay-out magnitude, whereas the information about the long-term valence was to be derived from a difference in likelihoods of observing that event in good or bad urns via inference. Therefore, a received reward can lower the long-term expected values and a received loss can increase long-term expected values via model-based inference, which we used to orthogonalise both dimensions of the pay-out events. Non-informative events still had a short-term valence (participants won or lost points), yet their likelihood-ratio between good or bad urns was always 1. On 10% of all trials, null-events were drawn that were non-informative and had a pay-out of 0 points.

Before each block, participants received detailed instructions about the distributions of all possible pay-outs in the upcoming block via pie charts (Fig. 1b) and participants were instructed to memorise these distributions as well as possible. However, to avoid misinterpretation of pay-outs, pie charts of the current pay-out distributions were continuously shown at the right side of the display during the whole experiment. Each block included 20 trials and the task consisted of 12 total blocks, resulting in 240 trials. In 10 blocks, one urn was good and the other one was bad, in two blocks both were good or bad, respectively. For all participants, the third block in the experiment consisted of two bad urns. Participants were informed that they could maximise their earnings by prioritising choices of good urns during the blocks, but additionally gain bonus money (5 points each) if they correctly identified the long-term valence of each urn at the end of a block in a gamble (incorrect identification resulted in a loss of 5 points, avoiding to guess did not change points). The exact expected value per block per urn was $+30$ and $-30$ points in good and bad urns, respectively, or, vice versa $\pm 1.5$ points per choice. The correct long-term assessment of an urn's valence (i.e., its long-term EV) was

additionally rewarded between blocks (Fig. 1b). Therefore, participants could earn up to $\pm 30$ points per block of 20 choices, while correct identification of each urn at the end of each block provided up to 10 additional points. During the task, the type of event (informative, non-informative, congruent or incongruent) was predetermined to keep the amount of information constant between all blocks and participants. Thus, each trial was observed with a frequency matching its exact probability in a block indicated in the pie charts. However, the pay-out depended on participants' choices, such that a congruent event was associated with a positive pay-out if a good urn was chosen, and vice versa if a bad one was chosen. Overall, participants displayed exploitative behaviour (average number of choices of good urn $= 153 \pm 5$ (SE), number of choices of bad urn $87 \pm 5$, $t_{23} = 6.60$, $p = 9.75 \times 10^{-7}$). However, they still sampled from the bad urn repeatedly even if they estimated the other urn to be good, which can be interpreted as exploratory behaviour (average number of choices of bad urn in blocks with one good and one bad urn and belief of good urn $> 0.6 = 33 \pm 5$, $t$-test against zero $t_{23} = 6.49$, $p = 1.27 \times 10^{-6}$). This resulted in participants being equally well able to evaluate the good urns' valence (correct $= 92 \pm 2\%$) as well as the bad urns' valence (correct $= 86\% \pm 5$, for difference $t_{23} = 1.45$ p $= 0.16$) at the prompt between blocks (they chose not to guess at all, a third option, in $6\% \pm 3$). On average, participants earned $174 \pm 13$ points in the task (range: 80–280), out of which 67 ($\pm 14$) were earned via exploitation within blocks, and 107 ($\pm 4$) were earned by correctly estimating urns' long-term valences during blocks as bonus pay-out. Prior to scanning, all participants performed a training block of 20 trials after which they could discuss possible questions with the experimenter.

**Computational modelling**. We first constructed an iterative Bayesian model that learned the task in an ideal, or normative[10], way, reflecting unbiased model-based learning. The model calculated the posterior belief ($B_{t+1}$) of an urn to be good based on the prior belief ($B_t$) associated with this urn and the conditional probability of the observed event ($E_t$) under the provided pay-out distribution in a good urn, scaled by the overall likelihood of observing this event incorporating the complementary belief ($1-B_t$), that is the belief that the urn was indeed bad, and the provided pay-out distribution in these respective urns:

$$B_{t+1} = \frac{P(E_t|\text{Good}) \times B_t}{P(E_t|\text{Good}) \times B_t + P(E_t|\text{Good}) \times (1-B_t)}$$

Such iterative Bayesian models have been demonstrated to match human behaviour in multiple tasks well[10,14,46]. Application of a delta rule leads to a measure of signed belief update ($\Delta B_t$) per event:

$$\Delta B_t = |B_{t+1} - B_t|$$

The Kullback–Leibler divergence ($D_{KL}$)[16] reflects the absolute change in Bayesian belief, or how much the model was updated per observation:

$$D_{KL} = |B_{t+1} - B_t|$$

Please not that this $D_{KL}$ derived from Bayesian updating is identical to a symmetrical $D_{KL}$[47]. Please furthermore note that in our task, $D_{KL}$ is not confounded with surprise[48], or Shannon-Information[16], as the only events that could be expected with higher probability than all other possible outcomes, were informative and thus led to model update, whereas non-informative events never led to model update, but were more rare, i.e., surprising. Furthermore, as we additionally wanted to test if subjective model update was influenced by model-free learning and investigate its neural correlates, we also applied the normative model to each participant's prompted prior beliefs (which will be called subjective Bayesian model below).

Next, to assess model-free learning, we used a simple Rescorla-Wagner based reinforcement learning model[15] to estimate single trial reward-prediction errors ($\text{RPE}_t = \delta_t$) over the course of task progression based on the discrepancy between the experienced pay-out and the expected value of the outcome on each trial ($EV_t$):

$$\delta_t = p_t - EV_t$$

All pay-outs were scaled to the highest possible outcome per block to account for likely range adaptation effects[49,50] and the expected value was updated with the trial-wise prediction-error scaled by a fixed learning rate ($\alpha$):

$$EV_{t+1} = EV_t + \alpha \times \delta_t$$

Based on each participant's sequence of choices, we minimised the log distance between the models expected value and each participants' subjective belief ($B_{subj}$) using maximum likelihood estimation (MLE) of the only one free parameter ($\alpha$):

$$LD = \sum_{t=1}^{T} \log(1 - |B_{subj} - EV_t|)$$

Resulting mean MLE $\alpha$ values were 0.0613 ($\pm 0.013$) across participants, indicating that slow integration of outcomes, reflected in a relatively low learning rate, was required in the task to match observed beliefs of the participants. This is plausible given the large discrepancy between an urns long-term expected value and the outcome magnitude of each trial (long-term expected values were always $\pm$ 30 points per block, whereas each trial's pay-out could be $\pm$ 60 points). Please note that we fit models to beliefs, rather than choices, as beliefs mediated choices in our

task (Supplementary Fig. 2) and we intentionally decoupled choices by encouraging explorative behaviour.

**Behavioural regression model.** For all comparisons of model predictions, we conducted multiple linear regression analyses on each individual participant's normalised belief updates (calculated as the difference between posterior and prior belief prompt), always accounting for edge (beliefs = 0 or 100) as a factor of no interest and additionally normalising model predictions. Resulting regression weights were then tested against zero at group level. First, comparison of a model including only $RPE_t$ as an additional predictor (negative log likelihood (−LL) = 8.029) and a model including normative Bayesian update (−LL = 6.611), revealed a much better fit for the Bayesian model (difference of summed −LL scores 2.837, likelihood-ratio test $p = 0$ within machine precision). However, both factors exerted a significant positive effect on participants' belief updating (both $p < 10^{-5}$). For the final model (Fig. 2d), we used the subjective Bayesian model described above from which we also derived predictions for the fMRI analysis ($D_{KL}$, $\Delta B_t$) as it showed the best fit (−LL difference to normative Bayesian model 242, likelihood-ratio test $p = 0$) and also likely fits neural data better in cases when participants incorrectly updated their beliefs (due to model-free biases or otherwise). Similar results as reported in the manuscript were, obtained when we compared normative Bayesian and RL models' predictions, or standardised regression coefficients instead of normalised predictors. Furthermore, the influence of RPE on participants' beliefs was mainly driven by the outcome component of the RPE and we found very similar results when we replaced the RPE regressor with the signed outcome per trial ($t_{23} = 3.76$, $p = 0.001$, compare to Fig. 2d).

**MRI data acquisition and analysis.** Functional MRI data were acquired on a 3 T Siemens TRIO scanner and pre-processing was carried out in FSL[51]. The task was presented using Presentation (Neurobehavioral Systems). For fMRI recording, an isotropic resolution of 3 mm was used with a repetition time (TR) = 2 s, echo time (TE) = 30 ms, and a flip angle of 80°. The number of volumes acquired was dependent on participants' behaviour and the mean number was 1549 (range 1352–1790) resulting in an average task duration of 52 min. Field maps were acquired using dual-echo gradient echo sequences with echoes at 4.92 ms and 7.38 ms using a repetition time of 600 ms and a voxel size of $1.9 \times 1.9 \times 3.0$ mm$^3$ in a grid of $240 \times 240 \times 112$ mm$^3$. Structural T1 images were acquired using an MPRAGE sequence with 1 mm isotropic resolution, TR = 2500 ms, TE = 4.77 ms, inversion time (TI) = 1100 ms, and a grid of $256 \times 256 \times 192$ mm$^3$.

fMRI data were motion corrected using rigid-body registration to the central volume[52] and aligned to the structural images warped into MNI space using affine registration[53], while applying field map based geometric undistortion. Low-frequency shifts were removed using a 100 s high-pass filter and slice time acquisition differences were corrected using Hanning windowed sinc interpolation. A Gaussian filter with 5 mm full width at half maximum was applied for spatial smoothing. To account for temporal autocorrelation, the GLM was fit into pre-whitened data, and all regressors were convolved with the standard hemodynamic response gamma-function (SD = 3, mean lag = 6 s).

The GLM included three parametric regressors of interest consisting out of $D_{KL}$, $RPE_t$, and $\Delta B_t$ at time of outcome presentation, which were derived from the computational models described above. Additional regressors modelled the main effect of outcome onset, the onset of the following belief prompt, and the response given prior to feedback, as well as the motion parameters from the motion correction. If not stated otherwise, all reported results are thresholded at $p < 0.001$ cluster-based correction for multiple comparisons with a cluster-extent threshold of $p < 0.05$. Conjunction analyses reflect regions that passed this threshold in each constituent analysis with effects of the same sign.

Across participants, effects were tested using a second level model including for each participant the behavioural regression weights for the influence of the Bayesian and RL models, as well as the intercept on the contrast of coefficient estimates between $D_{KL}$ and RPE. We then calculated the contrast for Bayesian influence over RL learning to identify brain regions where the relative degree of coding model-based and model-free update covaried with the relative influence of Bayesian updating compared to model-free biases across participants. In order to determine whether this effect was driven by increased RPE or $D_{KL}$ representations, we followed this analysis up by comparing the effect of individual results for $D_{KL}$ or RPE instead of their contrasts within regions found in the first analysis (Supplementary Fig. 4a, b). Similarly, we assessed the same question on the behavioural side by conducting a control analysis using each behavioural regression weight in a separate analysis (Supplementary Fig. 4c).

Time courses for regression analyses were derived from individual participants from a region-of-interest analysis from MNI coordinates placed along a dorsal to ventral gradient in the striatum with 5 mm distance between each other (Fig. 3b), or reflecting peak activity of the respective main effects in the other analyses (Supplementary Fig. 3c–h). Activity of one voxel per analysis was extracted and transformed to each individual participant's space by using the same registration as in the whole brain analyses. Extracted time courses of 10 s duration were locked to payout onset and oversampled by a factor of 10 and multiple regression was then applied to every pseudo-sampled time point separately. Therefore, time courses reflected averaged beta weights across participants (shaded areas = SE). Regression

models for the gradient analysis included the same control regressors as the whole brain GLM, and all other models as described in the corresponding sections.

For the analysis of single-trial Bayesianness of updating, we calculated Bayesianness as:

$$\text{Bayesianness}_t = 1 - |\Delta B_t - \Delta \text{IBU}_t|$$

where $\Delta B_t$ reflects each trial's ideal update derived from the Bayesian model (see above) and $\Delta \text{IBU}_t$ is the difference between a participants prior and prompted posterior belief on each trial (IBU individual belief update).

For the across-participants gradient analysis, we derived the slope (beta weight) from each participants' regression analysis of beta weights for the respective factors ($D_{KL}$, RPE) against the z-axis (ventral negative, dorsal positive), averaged over left and right hemisphere. These were then predicted in a multiple robust regression model by the behavioural regression weights for Bayesian updating, RL influence, and the intercept of the behavioural model.

**Data availability.** All data and code is available from the authors upon reasonable request.

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

## Acknowledgements

We thank Claus Tempelmann for help with fMRI scanning, Christian Kaufmann for help with fMRI analyses, as well as Renate Blobel and Denise Scheermann. For data acquisition of the replication study, we thank Cindy Lübeck, Christina Becker, Laura Waite and Yan Arnold for their support. Furthermore, we thank Anne Corcos, Hadriën Orvoën, and Gerhard Jocham who provided valuable input to the analyses, as well as Tanja Endrass for helpful comments on the manuscript. A.G.F. is funded by CBBS ScienceCampus financed by the Leibniz Association (SAS-2015-LIN-LWC) and S.B.G. supported by NESSHI, AAP Open Research Area in Europe.

## Author contributions

All authors conceived the study goal and design, interpreted the results, and wrote the manuscript. A.G.F. and M.U. developed the strategy of main and control analyses. A.G.F. programmed the task, collected and analysed the data.

## Additional information

**Competing interests:** The authors declare no competing financial interests.

