## [Peer Review File · Nature Communications]

Reviewers' comments:

Reviewer #1 (Remarks to the Author):

This paper describes an fMRI experiment using an elegant novel paradigm to dissociate different types of learning. In the task, participants must learn which of two urns has the higher overall payout. Because each urn has a known payout distribution, payout on an individual trial can be informative about the urn's identity – and positive payout on a single trial can indicate the negative urn overall.

Behaviourally, subjects learn using knowledge of the payout structure but are also influenced (sub-optimally) by short term reward prediction errors.

The authors use fMRI to show dissociate unsigned belief updates (KL divergence), about the urn's long term value, based on knowledge of the payout structure from signed belief updates based on the immediate reward prediction error which do not indicate the urn's long term value. They show that the dorsal striatum, ACC and parietal cortex are activated by the former whilst ventral striatum and frontal pole are activated by the latter. There is a dorsal-ventral gradient in the striatum and the distribution of activity along this gradient relates to individual differences in how much participants are biased by short term outcomes.

I think this is a very clever new paradigm and the results are interesting. On the other hand I think the contrast between two learning types conflates a few issues which could perhaps be teased apart to further improve the paper, and the chosen framing of long-term vs short-term doesn't capture the key difference between the types of learning as well as some other terminology might.

Throughout the two types of prediction are referred to as 'long term' and 'short term' but they may at least as justifiably be called 'model based' and 'model free' or 'unsigned' and 'signed'. Which is the more important factor?

Signed vs unsigned

It would be possible to test for effects of signed Bayesian model update and unsigned RPE from the RL model – in fact the former is included in the GLM for the fMRI analysis. Do the areas concerned with DKL also respond to unsigned RPE, and the areas concerned with signed RPE also respond to signed Bayesian update?

Long term vs short term

It seems that the learning rate for the RL model is very low (p13) – is it really correct to refer to the effect of RPE as a short term representation then, since it is taken with reference to a prediction that develops over a long time frame?

Model based vs model free

I think perhaps this is the key distinction (if the authors can show the effect is not explained away by the signed/unsigned distinction) – this seems to be an elegant new task for dissociating the two types of learning.

I would be inclined to frame the paper in terms of updating the generative model (which is the good urn?) using knowledge of the structure of the environment, vs model-free updating – providing that the issue of signed vs unsigned update can be shown not to be a confound.

Minor

The equation for the KL divergence gives DKL as the absolute difference between the probability that the blue urn is the good urn at trial t and trial t+1. This seems quite different from the formula I am used to seeing (on Wikipedia!), which I think in this case would be

$$DKL = B(t+1)[\log(B(t+1)) - \log(B(t))]$$

Could the authors please explain how they arrive at their formula? Or, should this quantity be referred to as unsigned model-based belief update rather than DKL?

Reviewer #2 (Remarks to the Author):

The authors aim to study whether short-term outcomes bias behaviour independent of long-term expectations based on an explicit model of the environment. I find the task very elegant, where the same value outcome may signal whether that choice has a positive or negative expected value when chosen repeatedly. The authors then show that prediction errors in both short term outcomes and long-term expected values drive belief updating when measuring explicit reported beliefs. The authors then show that long-term model-based prediction errors are represented more in the dorsal striatum and ACC, while ventral striatum represents model-free prediction errors. This finding is novel in the sense that previous studies using different tasks of model-based and model-free behaviour (e.g. Daw et al. 2011) showed only ventral striatal effects of model-based and model-free prediction errors. I particularly like that there is no uncertainty about the model of the environment, in that the subjects are continuously reminded of the possible probability distributions of the urns, and that immediate reward is uncoupled from the long-term expected value.

However, I have a number of comments and recommendations that need to be addressed.

First, I think the abstract and introduction could be rewritten to appeal to a more general audience, be more specific and clear about the question, and use less jargon. For example:

The first line of the abstract suggests that the task might relate to some sort of trade-off between short and long-term benefits of a choice (e.g. temporal discounting), when actually the paper and task are about how a single outcome may convey both short-term gains/losses and long-term expected value, but there is no trade-off between a long-term and short-term option as such. It would be good to make this very clear.

Overall, the abstract and introduction are very abstract, should be re-written for more

general audience, with less detail/jargon and more about the implications of the findings. For example, the authors use terms like 'model-free vs model-based evaluation', and 'the relative influence of information over reward', 'a priori full model knowledge', 'belief update' which are all quite jargony terms.

The sentence "choose actions purely based on beliefs, which does not require prior experience and can sometimes lead to decision against pleasurable prior expectancies. " is again very abstract: what do you mean by not requiring prior experience? Experience with what? And where does the belief arise from? And again "monkeys do not seem capable to process information independently." Independently of what? Furthermore, the claim that the 'ability to down-weight short-term outcomes... contributing substantially to the evolutionary success of the human species' is rather a big one, not backed up with arguments, and not really necessary for the paper.

I also have a number of concerns regarding the task, and modelling of the behavioural data:

1. Within the blocks of 20 trials, marbles were drawn without replacement (i.e. the proportions were exactly the same for all blocks). Why was this the case? This makes the relationship of the short-term outcome and long-term expected value more complex, as the long-term expected value is the average of all the marbles that have not yet been drawn. Is there any evidence that subjects used this information to adapt their strategy, i.e. pick the bad urn when they had 'used up' all the good options in the good urn? To test this, the authors could use the actual expected value (i.e. the mean of all the 'marbles' that are 'left in the urn' still), given the marbles that have already been observed in a block.

2. My key concern is that the authors model the beliefs of the subjects, rather than the subjects' choices. Why is this? We have long known in psychology that introspection is certainly by no means an unbiased window on what drives our choices. Presumably the ultimate aim is to understand the subjects' behaviour, not their estimates of the current state (or at least not the explicit reported estimates, but rather the implicit estimates that drive choice). This makes the current paper very difficult to compare to any other papers that talk about model-based and model-free choice. Why not at least include an analysis of the choices as well as the explicit reported beliefs (and the associated neural effects?). How do the reported beliefs link to the explicit choices? Do you think that asking subjects to report explicit beliefs changes their choice behaviour?

3. The authors contrast a model-based and model-free learning strategy, but also conflate this with a Bayesian and an RL framework, which is unnecessary and may be confusing, particularly for readers not very familiar with computational modelling. This choice has as a consequence that models are hard to compare because the Bayesian model is normative (no free parameters), whereas the RL model needs model-fitting (of the learning rate). Why do you not simply use a Bayesian 'model-free' model, i.e. a normative model that does not incorporate full knowledge of the environment (i.e. the pie charts), to contrast with the Bayesian model-based model?

Furthermore, in order to assess the relative and independent contribution of both model-

based and model-free learning the authors use a multiple regression model that is partly circular: first, you fit a model using a learning rate, on the beliefs, and then you do a regression with prediction errors computed using the same learning rates.

Results

Minor points

p3 - 'likelihood ration of informative events remained constant over blocks - unclear what informative events refers to at this point.

make clear that in any given block, also two good or two bad urns could be present.

add how you determined that the excluded subject did not understand task instructions

p 11: nr of choices: that is the +/-? SD? range?

'n choices' - should this be filled out?

modelling: all payouts were scaled to the highest possible outcome per block - why?
Why did you not fit the choices, but rather the subjective beliefs? Choice and subjective belief are probably a readout of the internal belief state.

rephrase: p 10: perceived reward/punishment  received
change 'beliefs in the expected value' to 'the expected value

Reviewer #3 (Remarks to the Author):

The authors manuscript is a functional imaging study of an inference and learning task in humans, in which participants need to learn whether options are good or bad by integrating experiences in an appropriate way, and by avoiding inappropriate biases/use of information.

Overall, the manuscript is a very nice piece of work containing a very carefully analyzed data set, with a lot of thought going into understanding the cognitive processes and neural mechanisms underlying it. However, I believe that some of carefulness and the novelty is obstructed by aspects of the framing and the brevity of the interpretation/discussion section. In other words, by emphasizing specific novel parts of the results in the main manuscript and supplements more and focusing less on the already previously described aspects or interpretations (such us simple model-based vs model-free), I think, the manuscript could be significantly improved. Also, there is one particularly pertinent study missing in discussion or introduction (Scholl et al., 2015 JoN), which also looked at reward induced biases against long term, more optimal learning and highlighted frontal pole and dACC (here referred to as MCC). The study is sufficiently different from the current manuscript as to not affect the novelty, but is very relevant, while also suggesting some

alternative explanations regarding some of the neural findings (see below). In short, if the authors emphasize more the idea of inferential processes and how such inferences interact with, other, lower order reinforcement learning and primary reward experiences, I think they have very exciting manuscript (to put it bluntly it explains the difference between "this is what happened just now" (Good vs Bad, rewarding content) and "this is what it means" (Urn identity/Casual structure, informational content)). The distinction between experience and inference would also make a clearer clinical link.

Other than including more results in the main manuscript (e.g. the trial-wise Bayesianness is excellent!) and extending the discussion I only had a couple of comments regarding some of the descriptions of the findings and minor comments about labels, wording etc. . The fact my comments are long is a sign of my interest. It is my hope they will help the authors to make the novelty and importance of their findings clearer and will hopefully not be too much work, as I am not suggesting any large scale re-analysis.

Major Comments:

1. It would help immensely with the readers comprehension if the task description was way earlier. I had a hard time understanding why the authors could dissociate the things they could from the text. If they had included a simple, common sense, description of the task early, this would have been easily avoided.

E.g." We designed a simple binary decision task in which participants had to learn about two different urns in a blockwise fashion. However, as there was a more complex probability distribution of outcome magnitudes, both negative and positive, we were able to contrast the Informational content of a trial from its current rewarding nature. Simply put, a specific rewarding outcome could be more frequent in the bad urn leading both to a positive prediction error for the current outcome and an inferential (potentially bayesian) process of thinking it more likely it is the bad option. In other words, while participants need to process positive reward information they also need to assess the informational content of the reward, which might lead them to make a negative conclusion." Also Related to clarity, maybe explain distinction between informative and non-informative events earlier. Maybe even a figure/Illustration in the main manuscript about inference, specific events and learning might be nice. Similarly, some of the points below might make it necessary to move some of the analysis into main figures, such as Figure S5.

2. As mentioned above, I think Scholl et al. 2015 in JoN is a very relevant paper and should be discussed, as it showed that in a learning task when receiving irrelevant reward information, the effect-size of that reward signal at outcome in frontal pole and in their case dACC/MCC was predictive of how unbiased participants were. The authors of that study however interpret their effect in terms over the signal being necessary in order to overcome, suppress or compensate for such biases, rather than integrate it (see : "integration of model-free signals within regions coding Bayesian model update.").

Furthermore, in the Scholl study the authors regress against the outcome not a RPE. Seeing that RPE's are strongly driven by outcome signals, how much of the irrelevant/model-free reward signal preventing biased behaviour is driven by the outcome? I do not expect the authors to rerun all their analysis just with outcome, rather than RPE, particularly as they show some of the regions to have expectation suppression, but was curious what their

sense was, in case they looked at the difference at some point. Also, even if it was mostly about contextualizing reward experiences rather than a “real” RPE signal for learning, the process should get easier with repetition, giving a RPE like effect of reduced signal with repetition, without actually representing an RPE-like learning mechanism. In short, have a proper discussion of Scholl et al., and the implications (i.e. suppressing irrelevant information) and maybe briefly mention RPE vs pure outcome processing.

3. Putting expectation and outcome into a regression is very nice and beyond the level of proof most other learning papers offer! However, why the expectation is based on the Bayesian learner needs to be discussed more. See: “the model-free single-trial RPE most likely based on expected value estimated from model-based beliefs”.

If I am not mistaken this makes what they call the model-free RPE not actually model free, as the expectation is based on the optimal Bayesian belief structure. This has implication for their framing. More specifically, it implies that although vSTR does process primary reward information, it is nonetheless controlled/modulated by expectation that includes higher order beliefs. It also suggests that even though the region does not represent higher order reward signals such as odds of being in the good urn, it does modulate its expectation of something good happening by whether the option is seen to be the good or bad one according to the inferential process, making it sensitive to more “cognitive” inferential modulation.

If the authors discuss this bit more their nice differentiation between different expectation suppressions in Figure S2 might be nice in the main text. It could also require a discussion of why DKL regions don’t hold previous belief but only update. In this context it might be nice to cite Boorman 2016 in Neuron, as they found IOFC signals of updates but not representation.

4. Related to point 2, a lot of Model-based vs model-free papers ignore the fact that in order to be model-based a person needs to potentially suppress the reward experience (e.g. Daw’s famous two step task). In other words, rather than just talking about model-based use of transition probabilities, I think it is important to highlight in this manuscript and traditional model-based vs free tasks the fact that the cognitive phenomenon of suppression of reward information also exists. I think this might be important for understanding particularly the effects in this study of activity in frontal pole and other places that is both very Bayesian and activity that is strong when reward experiences are strong, as this is when interference needs to be avoided. Related to this, while the authors of this manuscript say DKL + RPE in dorsal striatum makes participants unbiased because of “integration” into learning, it could be precisely because of the opposite of not including it into a value estimate, but keeping the interference lower in the value learning system. Related to this there might be a title that includes the possibility of such a mechanism explaining the co-occurrence of long term/inference based value and short-term/reward experience based representations. [As a side thought, it ultimately might be interesting to measure physiological inhibition effects with e.g. MRS and see whether inhibition in target area correlates with less interference].

5. I think the authors could make a bigger deal about Figure S5, as they can only look at it

because they have the nice trial-wise quantitative rating after every trial. It is a very interesting result showing that if participants in a trial, manage to quantitatively update in a more Bayesian way, the dACC/MCC is engaged together with a network of other regions, such as dmPFC/medial Frontal Pole, posterior cingulate etc.. Many of those regions are interesting for people looking at dynamic decision making and learning and will be interested in this finding.

6. While many papers have talked about model-based and free and Scholl et al, talked about reward related biases and overcoming them, the striatal results are very carefully done and novel. The fact that the striatum isn't a unitary structure is very cool, but I think it might be nice if the authors spent a bit more time to relate their finding a little bit more to other ideas of dissociation between dorsal and ventral e.g. Trevor Robbins' work on compulsivity and habits and the ventral to dorsal transition. It wasn't clear to me how to square Robbins results with the current finding, but it is a well-known theory related to learning and it might be good if the authors said how they might relate.

7. I realize this might be more of a personal grudge of mine, but I find the term "model-based" terribly generic to the point of being meaningless. Even worse, it often obscures the novelty of a finding and does not highlight the functional significance enough. For example here, model-based inference relies on proper contextualization of a rewarding outcome, suppressing of the immediate reinforcing properties of reward and a statistical inference of an observation, while other "model-based" tasks such as Doll et al 2016 Nat Neuro, is called model-based but there model-based means generalization between two stimuli which both lead to the same outcome.

8. Were there any relevant/interesting decision-related signals in the task? I do believe the authors dissociated the outcome from decision phase temporally, but do not mention whether there were any interesting effects related to those analyses. E.g. was there an effect of last trial update on the decision phase? Or alternatively, was there a larger FPole signal when making decisions and having to go against a last trial rewarding, but ultimately/statistically bad urn?

Minor Comments:

A) Page5 "assuming gradual change (Fig3D)" gradients are very hard to prove, particularly in fMRI, as gradients could be an artifact of movement combined with discrete clusters. I don't think a gradient is particularly important for their argument, but I would just recommend a bit of caution about gradients.

B) The figure legends in Figure S1 seem off/incorrect. E.g. in C the p values are tricky to read. One says $p < 10$ with the -3 below the 10.

C) More labeling of figures might be nice. For example labeling the colours in figure 3 in the figure itself might be nice.

D) When the authors say "Understand how both learning mechanisms are implemented in

the brain." The sentence is slightly confusing as it is hard to reconstruct what both refers to.

E) The statement "Bayesian solution to the problem which was easily accessible to participants" is quite a strong claim. It is true that the statistical information was always present but explicit Bayesian solutions are rarely trivial. Maybe rephrase and say, "the information for Bayesian statistical inference was readily accessible and participants seemed to be guided by Bayesian like updating"

F) Is former later wrong way around in the abstract or are the authors saying short term in dmSTR and FPole and long term in vSTR?

G) The authors say "Which model applies to current context" which suggests model switching. It might be nice if the authors elaborated a little bit more, as they might also have meant which urn/ statistical environment applies instead of switching "models".

H) "Negative covariation between the behavioural influence of model-free prediction and the steepness of the RPE gradient" could be maybe said with viewer double negations. Took me a while to understand.

I) I Didn't find it needed the IGT discussion very much, as it is a rather outdated experiment. I however understand if the authors want to keep it.

J) It might be better to discuss the frontal pole subregions and general overlap in an anatomical region qualitatively rather than focus on the very few voxels that survive formal overlap analysis, but again, I defer to the authors preference on this.

To summarize, I congratulate the authors on a very nice study. With some reframing of the findings and inclusion of some more of the already done analysis in the main manuscript and some intuitive illustrations (as described in my major points below), I think this study can be very interesting for cognitive neuroscientists as well as a wider ranging community including theoretician and clinical researchers.

Reviewer #4 (Remarks to the Author):

SUMMARY

Fischer et al., aimed to investigate the differential effects on behaviour of learning from short-term (i.e., reward-punishment) versus long-term outcome information. Further, using fMRI, they intended to identify neural substrates of these different forms of learning. They measured participants' behavioural performance regarding judgments about the valence of long-term outcomes and how immediate/short-term feedback affected these judgments. They also measured neural activity related to receiving immediate/short-term feedback using fMRI.

Blocks

Participants performed 12 blocks with 20 trials per block. Each block was independent of other blocks in terms of the association of blue or yellow urns with good or bad long-term outcomes. On 10 of 12 blocks, one urn was associated in the long-term with a good outcome whereas the other was associated in the long-term with a bad outcome. In 1 block (always Block 3), both urns were associated with negative long-term outcomes. In 1 block, both urns were associated with positive outcomes. Inclusion of blocks in which both urns were associated with the same long-term outcomes was intended to encourage participants to select/sample both urns even after they had determined that one urn was good or bad, not being able to infer the long-term consequence of the other urn based on this information.

Trials

On each trial, in each block, a) a blue and a yellow urn were presented, randomly to the right or the left of fixation, b) participants selected one urn, c) an immediate (short-term) positive or negative feedback sum was presented, and d) participants were asked to indicate whether the urn was associated with a long-term good or bad outcome by increasing or decreasing a marker in units of 10 from 0-100. At the beginning of each block, the marker for each urn was set at 50%. Within each block, the adjusted position of markers associated with each urn was preserved across trials. In this way, the cumulative information across trials regarding each urn's association with positive and negative long-term outcomes was visually represented explicitly.

Long-term versus immediate outcome information

Prior to each block, the conditional probabilities of different immediate positive and negative feedback sums being associated with a good or bad long-term consequence were presented explicitly. To reduce working memory load, a pie-chart representing these conditional probabilities of different positive and negative feedback sums being associated with good versus bad long-term consequences was consistently displayed throughout the block. Participants were instructed to make their judgments about good or bad long-term consequences associated with the urn that they selected based on these a priori probabilities ascribed to immediate feedback sums. In this way, judgment about long-term consequences associated with the selected urn was appropriately achieved by cross-referencing the positive or negative feedback sum with the pie-chart that was continuously presented throughout the block. The selected urn's association with a long-term good or bad consequence was not directly related to the valence or magnitude of the immediate reward (i.e., positive feedback sum) or punishment (i.e., negative feedback sum). Some negative immediate feedback sums were more likely associated with a long-term bad consequence whereas other negative immediate feedback sums were more likely associated with a long-term good consequence. Conversely, some positive immediate feedback sums were associated with a long-term good consequence whereas other positive immediate feedback sums were more likely associated with a long-term bad consequence. Dissociating immediate and long-term outcomes created the possibility of looking at congruent versus incongruent cases to better assess the differential contributions of each type of consequence. On some trials, the immediate negative or positive feedback sum was not presented on the pie-chart and therefore these trials were uninformative of the long-term

association. This presented an opportunity for investigating the effect of immediate consequences in an unopposed way on judgments. Again, this should have no impact on updating of an urn's long-term outcome. On 10% of trials, a null event was presented in that it was not informative but resulted in 0 points won or lost.

Earning/losing points

Participants earned/lost points that were converted to monetary reward, based on the immediate feedback sums (up to 20 per block). They also earned 10 bonus points for correctly assigning the long-term good or bad outcome associated with each of the urns at the end of each block. Confusingly, though they were encouraged to explore both urns to be able to earn the 10 bonus points, they were also advised to 'maximize their earnings by prioritizing good urns'.

Modelling

A model of a Bayesian learner was created to represent performance based on assigning good or bad long-term consequences uniquely with reference to the probability pie-chart. This is the ideal performance, as valence or magnitude of immediate punishment or reward should not influence good or bad long-term consequence judgments based on instructions to participants. ΔB_t represented the sign (+ or -) of the belief update whereas DKL reflected the absolute belief update.

A separate reinforcement learning (RL) model was created to represent performance based uniquely on assigning good or bad long-term consequences on the basis of valence and magnitude of immediate feedback. Updating in this model was based on reward prediction error (RPE). That is degree of change given the current immediate feedback, relative to values of immediate feedback of trials that came before. In this model, behaviour was not influenced by accrual of information related to the long-term consequence associated with each urn.

Proposed Analyses

Behavioural analyses: The authors proposed that by comparing actual performance to each of these models, they could determine whether long-term models or immediate rewards or punishments were more influential on actual behaviour.

fMRI Analyses: They investigated BOLD signal at the time of immediate feedback, correlating with the amount of updating expected by the Bayesian Learner in a model that takes into account signed belief update (ΔB_t) or absolute belief update (DKL) versus the RPE expected based on RL model. They further investigated BOLD signal correlation, across subjects, with the behavioural regression weights related to the Bayesian and RL models. They contrasted Bayesian influence vs. RL at the group level. Within brain regions identified in these contrasts, they investigated the effect of absolute belief updating by the Bayesian Learner (DKL) versus RPE. Averaged beta weights from ROIs along a DS-VS gradient were compared to time courses for regression. They derived the slope from each participants' regression analysis of beta weights from ROIs along a DS-VS gradient for Bayesian Learning and RL models against z-axis (VS negative, DS positive). These were then predicted in a multiple regression model by the behavioral regression weights for Bayesian updating, RL influence, and the intercept of the behavioral model.

Results

Actual behaviour was more consistent with the Bayesian Learner Models though there was statistically significant biasing from the RL model as well, particularly in the incongruent case.

fMRI analyses revealed that different brain regions covaried positively with Bayesian model updates (i.e., DS, IPS, dlPFC) and RPE (i.e., VS, vmPFC). However, conjunction analyses revealed overlap in medial striatum as well as lateral frontopolar cortex (FPC) and posterior cingulate cortex (PCC). At the group level, Beta weights of DS and VS ROIs chosen along the DS-VS gradient revealed that DS associates significantly with DKL (Bayesian updating), whereas VS covaries with RPE. Closer examination of VS signal relative to different components of the RPE (i.e., immediate feedback/actual payout, expectancy based on RL model, and participant's actual belief about the longterm consequence of the urn on a given trial), they found that VS signal correlated with immediate feedback (i.e., actual payout) but also was negatively associated with participant's prior belief about the urn's valence in the longterm, reflecting intertwined processing of RL and Bayesian model learning in VS. Taking an individual difference approach, they regressed onto each individual participants' weights of the Bayesian versus RL models on belief updating obtained from the behavioural regression analysis onto the contrasts of parameter coefficient estimates for DKL and RPE in whole brain analysis. Resulting parameter coefficients were then again contrasted (contrast of a contrast). They found that the lower the difference between coding Dkl and RPE in DS and FPC, the more optimal and less biased by short-term outcomes was performance. Finally, to follow up on this contrast of contrasts, the authors regressed individual weights of RL versus Bayesian learning model on the striatal gradients from VS to DS for Dkl and RPE. They found the more RPE signals were focussed ventrally and spared the DS, the more RPE biased belief updates.

REVIEW

Fischer et al., aimed to investigate an incredibly important topic. They intended to understand the relative influence on behaviour of learning from immediate rewards/punishments versus learning from long-term outcomes. Further, they aimed to distinguish the neural mechanisms underlying these different forms of learning. The task that they implemented to tease apart these processes was extremely complex and not easily accessible in its current presentation. A large number of fMRI analyses were conducted to carefully disentangle these different influences on behaviour. Despite the study aims, as detailed below, the current task failed to unambiguously contrast learning from decision making based on short-term versus long-term consequences. These results are open to an alternative interpretation that is equally or more plausible than the one provided.

Major Criticisms

1) This is an incredibly complicated task, with numerous components and important details, and though summarizing the task clearly and in a manner that is easily accessible is a challenge, it is essential for readers to be able to critically review the study and for the study to be replicated. I devoted considerable time to piecing the study together and have struggled to faithfully summarize all components here.

Overall, I would recommend a high-level overview of the task, followed by sections, separated by subheadings, emphasizing and clarifying the important features of the task and its implementation. Try to avoid 'vice-versa' and spell out the different conditions. Use consistent naming throughout as well to simplify.

'Participants were informed that they could maximize their earnings by prioritizing good urns.... the exact expected value per choice per urn was +1.5 and -1.5 points in good and bad urns, respectively.'

This is difficult to grasp on first pass and should be clarified further, with a specific example perhaps and by adding a label to specify this in the figure. Further, consistently naming conditions or outcomes could simplify. Elsewhere they refer to short-term or immediate feedback as 'payouts'. Here 'payout in points' per choice might be easier than 'exact expected value per choice'.

'Prior to scanning, all participants performed a training block of 20 trials after which they could discuss possible questions with the experimenter. During the task, the type of event (informative, non-informative, congruent or incongruent) was predetermined to keep the amount of information constant between all blocks and subjects. Thus, each trial was observed with a frequency matching its exact probability in a block indicated in the pie charts. However, the payout depended on participants' choices, such that a congruent event was associated with a positive payout if a good urn was chosen, and vice versa if a bad one was chosen'

Is this predetermined proportion of event type applicable only to the block of trials in the practice session? The paragraphing suggests it but within the paragraph the reference to 'all blocks' would suggest that this is referring to the experimental blocks perhaps as well as the practice block. Given the complexity of this experiment, avoiding anything, such as above, that adds ambiguity will be very important.

In addition to the proportions of event types, was the trial order pre-determined? Order will have an impact on how distinct RL and Bayesian Learner models develop over the block. Because urn choice was participant dependent, fully controlled trial orders, event-types could not be achieved.

2) In the design of the task and in the introduction to the topic, the authors confound learning and deciding on the basis of immediate versus long-term consequences. Even the real-world examples provided in the introduction confuse these concepts. Rather than presenting situations in which problem behaviour results from difficulty simultaneously learning from short- and long-term consequences, both examples provide evidence of deficiencies in prioritizing long-term consequences over short-term ones in making decisions that guide behaviour. The authors themselves state plainly that people 'know' both the short-term and long-term consequences of their behaviour but are unable to resist behaviour that has positive immediate but negative long-term consequences or conversely in enduring short-term losses to achieve long-term gains. In effect, they describe scenarios

that have generally been used to explore the delayed discounting effect in the study of decision making (Ainslie, 1975). Unfortunately, these confounds carry through from the introduction to the experimental task.

3) In the current task, as implemented, participants are asked to make decisions on the selected urn's likelihood of being associated with a long-term positive or negative consequence, referring to a pie-chart that explicitly presents these conditional probabilities and was visible throughout the experiment. Correctly performing these decisions requires no learning at all. In contrast, participants had to resist being influenced by the valence or magnitude of the immediate/short-term feedback in making their judgment of the long-term consequence associated with the selected urn on each trial. Participants could perform this task optimally and generate the correct long-term consequence without learning anything, simply by making decisions with reference to provided conditional probabilities. On each trial, participants indicated their judgment about the long-term consequence of the urn that they were evaluating, by moving a marker up or down in increments of 10 along a 0-100 continuum, with greater than 50 being 'good' and lower than 50 being 'bad'. At the beginning of each block, the marker for the blue and the marker for the yellow urns were set to 50. After each trial, participants moved the marker associated with the yellow or blue urn, depending on which urn had been selected for exploration, to reflect their decision regarding the probability that the urn was associated with a positive versus a negative consequence. Of importance, the position of the marker for each the blue and the yellow urn was conserved across trials. In this way, even to answer the bonus question correctly at the end of each block, no true trial-by-trial learning or creation of a long-term model was required regarding yellow or blue urn's 'good' or 'bad' status. The marker provided an explicit visual representation of the culmination of all judgments good or bad for each urn. To answer the bonus correctly, participants would have only to recall whether the marker was above or below 50 on the final trial to accurately perform the bonus question. Although some learning/internalization of a long-term model was not precluded, as the task was designed, learning this long-term outcome was not required to accurately perform. Consequently, the authors' contention that they were directly studying belief formation itself was not warranted and the interpretation of the fMRI data is compromised by this ambiguity. In the larger literatures, DS, IPS, and dlPFC, which were associated with Bayesian learning/belief formation model in fMRI analyses, are previously extensively linked to decision making, particularly when influences on selections are ambiguous. The Bayesian Learner model could equally have been called the Bayesian Decision model, revealing higher activity in brain regions with more optimal decision strategies that are freer from bias from irrelevant feedback. All analyses related to the Bayesian decision/learning parameters can be re-interpreted in this way. Of brain regions that preferentially correlate with the Bayesian model, only the DS is also frequently implicated in learning. Increasingly it is being suggested that DS actually mediates decisions but paradigms often confound learning and decision processes (Hiebert et al., 2014, Neuroimage; Atallah et al., 2007, Nature Neuroscience).

4) ΔB_t , change in signed belief, was calculated and used in behavioural analyses but then is no longer referenced in the fMRI analyses. Were the analyses with ΔB_t similar to those with DKI as would be expected? Does this parameter add anything further? Should it be

included?

5) The explicit instruction to participants is to sample/explore both urns to learn about good or bad long-term outcomes but also to prioritize selecting urns with long-term good consequences to maximize payouts. Because the positive immediate feedback sums were greater for good relative to bad urns, the finding that 153 good and 87 bad urns were sampled on average cannot be interpreted as evidence of learning the long-term outcome associations for each urn. This bias toward selecting good urns could have occurred because good urns on average payout +1.5 points in the immediate/short-term, whereas bad urns on average payout -1.5 points per choice. In this way, the RL model will be slightly biased to favour urns associated with good relative to bad long-term outcomes.

6) 'On average, subjects earned 174 ± 13 points in the task (range: 80 - 280), out of which 67 (± 14) were earned via exploitation within blocks, and 107 (± 4) were earned by correctly estimating urns' long-term valences during blocks as bonus payout.'

This indicates that participants earned more from bonus questions (potentially 10 per block) at the end of each block than from maximizing points throughout by choosing the urn with good long-term consequences (potential 20 per block). Because the urn with good long-term consequences at times is associated with negative immediate feedback sums, it's unclear how to interpret the number of points earned from participants' sampling behaviour. What was the total possible points that could be earned if participants performed in an ideal way, always sampling from the urn that was associated with positive long-term consequences? This will be difficult to determine because participants need to sample both urns at least on a few trials to determine the long-term consequence associated with each. How did sampling behaviour differ for blocks in which both urns were associated with good or with bad long-term outcomes, relative to blocks in which one urn was good and the other was bad in the long-term? Differences in sampling behaviour between these blocks could potentially provide some evidence of an internal model developing. No difference in sampling between these blocks would favour the view that these findings provide evidence for brain regions that underlie learning versus those that enact decisions.

5) At odds with the notion that greater DS activity was associated with Bayesian Decision/Learning, they found that the lower the difference between coding Dkl and RPE in DS and FPC, the more optimal was performance and the less biased it was by short-term outcomes. Similarly, to follow up on this contrast of contrasts, the authors regressed individual weights of RL versus Bayesian learning model on the striatal gradients from VS to DS for Dkl and RPE. They found the more RPE signals were focussed ventrally and spared the DS, the more RPE biased belief updates.

Could these findings have arisen from comparing cases in which the DKL and RPE were more similar to one another, relative to cases in which they differed to a greater extent? Was this possibility investigated and excluded? Because trial order was not predetermined and participants' urn selections were idiosyncratic, Bayesian models and RL models will differ for each participant and the difference between these models will also vary.

6) One subject was removed from the analysis for not understanding the task instructions. How was this determined? Was this participant removed before analyses were attempted?

7) There are some unreferenced statements, conclusions that are not well supported by data, or that extend findings too broadly.

Minor criticisms:

1) The authors claim that rodents' are unable to simultaneously learn long-term outcomes and disregard immediate rewards. This is presented as a motivation for performing the present imaging study. This statement is not backed up with references. Are there empirical demonstrations? Further, this claim would be at odds with their interpretations that learning from long-term outcomes is mediated by DS, IPS, PFC whereas learning from immediate rewards is mediated by VS, brain regions that have homology in rodents.

2) 'Finally, activity in dopaminergic midbrain regions likely including the ventral tegmental area (Fig. S5) covaried with ideal updating; thus,...'

This is the first time that authors mention VTA, not described in the results section or even in the figure that is referenced. Because of their proximity, VTA and adjacent SNc activations in the brainstem cannot be differentiated. Authors declare that this activity is most associated with ideal long-term model updating (i.e., learning of long-term consequences), but an equally apt interpretation is that this region correlates with ideal decision making with relation to the conditional probabilities. SNc nearly exclusively innervates DS, the brain region that was shown to most associate with Bayesian Model Learning (or equally Bayesian model Decision making) in this study. Further a large literature ascribes decision making/response selection with DS to provide context. Finally, VS, which is VTA-innervated, was most associated with RL model (RPE) and to a much lesser extent the Bayesian Learner model, to which the ideal updating refers.

3) 'This appears compatible with previous findings that activity in dopaminergic midbrain regions increases with task demands^{26,27}: it could be speculated that increased midbrain activity could support ventral-to-dorsal information transfer and integration within the striatum and increase gain in cognitive control areas.'

This was not demonstrated by the results. Correlations between midbrain regions, VS versus DS, and behavioural change were not investigated.

4) 'Perhaps this could help to explain the success of mindfulness-based therapies in various psychiatric fields including the therapy of addiction'

This extends the findings too broadly and is an unnecessary statement.

Response Letter

First of all, we thank the four reviewers for their insightful comments and suggestions for improvement of our manuscript. In our opinion, the adaptations in response to their comments helped to considerably improve the article and the suggested additional analyses extended its scope significantly.

As suggested by all reviewers, we have focussed the manuscript more strongly on the difference between model-free reward-based and inference-based learning. This is also reflected in the new title of the manuscript, which we changed to „How reward experience biases inference“.

In response to concerns of reviewer 4 related to the design of our study, we have modified the task design with respect to how and when we prompt participants' beliefs. We collected an additional fMRI dataset ($n = 18$) in which we replicate all main findings related to dissociable processes related to reward- and inference-based learning. On the one hand, this supports the interpretation of all results of the original dataset. On the other hand, we feel that a replication in an independent sample, that was additionally collected on a different MRI machine (Siemens Skyra), add value to the study itself, especially given the ongoing debate about reproducibility in science.

Furthermore, as suggested by reviewer 1, we clearly demonstrate that the effects of D_{KL} and RPE cannot be reduced to signed belief updating or unsigned reward prediction errors by including all factors into a control analysis (Figure R1). We additionally used logistic regression of choice behaviour, to demonstrate that prompted beliefs in our task truly are the main factor that drives choices.

In the following, we will comment on the points and arguments made by all reviewers, outline the changes we have made because of those suggestions, and offer our opinion on several points which we feel needed further clarification from our side. The reviewers' comments will be included in our response letter (in italic print for clarity) combined with the responses to the comments (in straight blue print). Unless otherwise specified, the page and figure numbers refer to the revised version of the manuscript and figures used only in the response letter are referred to with a capital R. We marked changes in the manuscript (other than simple spelling and Figure number changes) in blue.

Reviewers' comments:

Reviewer #1

This paper describes an fMRI experiment using an elegant novel paradigm to dissociate different types of learning. In the task, participants must learn which of two urns has the higher overall payout. Because each urn has a known payout distribution, payout on an individual trial can be informative about the urn's identity – and positive payout on a single trial can indicate the negative urn overall.

Behaviourally, subjects learn using knowledge of the payout structure but are also influenced (sub-optimally) by short term reward prediction errors.

The authors use fMRI to show dissociate unsigned belief updates (KL divergence), about the urn's long term value, based on knowledge of the payout structure from signed belief updates based on the immediate reward prediction error which do not indicate the urn's long term value. They show that the dorsal striatum, ACC and parietal cortex are activated by the former whilst ventral striatum and frontal pole are activated by the latter. There is a dorsal-ventral gradient in the striatum and the distribution of activity along this gradient relates to individual differences in how much participants are biased by short term outcomes.

I think this is a very clever new paradigm and the results are interesting. On the other hand I think the contrast between two learning types conflates a few issues which could perhaps be teased apart to further improve the paper, and the chosen framing of long-term vs short-term doesn't capture the key difference between the types of learning as well as some other terminology might.

We are very pleased with the reviewer's assessment of our paradigm and respond to the points raised by the reviewer in more detail below. In short, we have changed the terminology throughout the manuscript and now refer to the two types of learning as model-free reward learning / experience and inference-based learning, as this was also pointed out by reviewer #3. We believe that this terminology is more precise and provide an example to demonstrate that this translates and includes short- and long-term learning in some, but not all cases. Furthermore, we think that the type of learning required to solve the task here can be subsumed to be „model-based“ learning, but this is a very general term, as application of various kinds of models could be termed model-based learning (i.e., spatial, temporal or even moral models could be employed for learning, yet one would expect significant differences in their respective neural implementations). Therefore, we chose the term inference-based learning which we think more precisely describes learning in our task and we additionally changed the title of the manuscripts to reflect this.

Signed vs unsigned

It would be possible to test for effects of signed Bayesian model update and unsigned RPE from the RL model – in fact the former is included in the GLM for the fMRI analysis. Do the areas concerned with DKL also respond to unsigned RPE, and the areas concerned with signed RPE also respond to signed Bayesian update?

To address this very interesting question, we regressed *signed* D_{KL} as well as *unsigned* RPE onto the BOLD signal time course in the areas identified in the main analyses on whole brain level, in a model that otherwise included the same regressors as in the main analysis.

Firstly, we compare the effects of D_{KL} and unsigned RPE, that is the hypothesis that reward surprise in the form of an unsigned RPE could explain the effect of D_{KL} , or reflect something very similar. We find that the D_{KL} effect remains intact in all relevant regions identified in the main analyses (see Figure R1 below). Additionally, weak effects for unsigned RPE coding were seen in the left (peak $p = 0.021$), but not right dorsal striatum ($p = 0.085$), the dIPFC (peak $p = 0.003$) and IPS (peak $p = 0.0018$), but not aMCC (peak $p = 0.15$). However, in all regions the effect of D_{KL} was much stronger than the effect of unsigned surprise, excluding that D_{KL} can be reduced to unsigned learning signals.

Similarly, all main effects identified for regions that covaried with RPE were unchanged by inclusion of *signed* D_{KL} , as would be expected as this regressor was accounted for in the main GLM. Additionally, we saw a significant covariation between signed belief updating and the BOLD signal in the right accumbens ($p = 0.02$), indicating that this region reflected both short-term rewards and updating of long-term expectancies.

Figure R1: Comparison of D_{KL} and unsigned RPE (blue and green in top plots) and RPE and signed $D_{KL} / \Delta B_t$ (red and purple in bottom plots). All main effects of D_{KL} were seen over and above the effects of unsigned RPE signals (reward surprise) and the effects for RPE were seen over and above the effects of long-term valence updating defined as signed D_{KL} or ΔB_t . The right accumbens displayed covariation with long-term signed belief updating (bottom middle plot).

Long term vs short term

It seems that the learning rate for the RL model is very low (p13) – is it really correct to refer to the effect of RPE as a short term representation then, since it is taken with reference to a prediction that develops over a long time frame?

We agree with the reviewer that the term “short-term” does not fit exactly, only the outcome part of the RPE signal reflects short-term evaluation. The learning rate is very low, which is explained by the fact that in order to learn the task, an RL algorithm needs to collect many samples, which is an inefficient solution to the task. However, the RPE combines both, short-term outcome and (to an inefficient degree compared to the Bayesian model) long-term expectancy. In this case, when the learning rate is low, the covariation with brain signals is driven predominantly by the outcome part. This fits well to the results presented in Figure S3g,h, that show that the ventral striatum neither reflected expected values of the RL model, nor of an RL model with a shorter integration time (i.e., higher learning rate), yet did reflect beliefs. We discuss this in more detail in the revised discussion of the main manuscript on p. 10.

Model based vs model free

I think perhaps this is the key distinction (if the authors can show the effect is not explained away by the signed/unsigned distinction) – this seems to be an elegant new task for dissociating the two types of learning.

I would be inclined to frame the paper in terms of updating the generative model (which is the good urn?) using knowledge of the structure of the environment, vs model-free updating – providing that the issue of signed vs unsigned update can be shown not to be a confound.

We thank the reviewer for this suggestion and think it is well compatible with the suggestions of reviewer #3 as well. As the signed vs unsigned comparison did not explain the pattern of results we found (see response above), we have rephrased the terminology of the paper. We refer to updating of the generative model as inference, which we hope that the reviewer agrees with us, nicely links to the Bayesian framework we use and is a more precise terminology than the broader framework of model-based learning, in which the actual model may be manifold.

Minor

The equation for the KL divergence gives D_{KL} as the absolute difference between the probability that the blue urn is the good urn at trial t and trial $t+1$. This seems quite different from the formula I am used to seeing (on Wikipedia!), which I think in this case would be

$$DKL = B(t+1)[\log(B(t+1)) - \log(B(t))]$$

Could the authors please explain how they arrive at their formula? Or, should this quantity be referred to as unsigned model-based belief update rather than D_{KL} ?

The reviewer is right that our terminology with D_{KL} is slightly imprecise, and we acknowledge this in the methods section of the revised manuscript. The Bayesian formula we use reflects a symmetrical variant of D_{KL} (see Johnson, D., & Sinanovic, S. (2001). Symmetrizing the Kullback-Leibler Distance. IEEE Transactions on Information Theory for details), which is defined as

$$[D_{KL}(P_1, P_2) + D_{KL}(P_2, P_1)] / 2$$

and

$$D_{KL} = \sum P_i \times \log_2 (P_i / Q_i)$$

We use this as we feel that an *asymmetric* D_{KL} , while fully valid in information theory, may be difficult to grasp for readers not that acquainted with its subtleties, yet that D_{KL} itself is a well known measure. The asymmetric D_{KL} would additionally be almost perfectly correlated with the (Bayesian) D_{KL} implementation we used (empirically, the average r between both measures across the sample is 0.997), such that all results would be very similar.

For a discrete example, consider in the current experiment updating a prior belief of 0.8 that an urn is good (0.2 that it is bad) to 0.9 that it is good (and 0.1 that it is bad). This would be reflected as $D_{KL} = 0.064$. The inverse change from 0.9 to 0.8, however, is measured as a D_{KL} of 0.053, displaying the asymmetric property of D_{KL} . The symmetrical measure defined above would be 0.0585, which is fully proportional to the Bayesian D_{KL} term we use.

Note that D_{KL} is only defined in a non-negative range, therefore, we would rather prefer to remain with the current terminology and differentiate a *signed/directed belief update* from the (in this sense unsigned) D_{KL} , which cannot be negative.

Reviewer #2 (Remarks to the Author):

The authors aim to study whether short-term outcomes bias behaviour independent of long-term expectations based on an explicit model of the environment. I find the task very elegant, where the same value outcome may signal whether that choice has a positive or negative expected value when chosen repeatedly. The authors then show that prediction errors in both short term outcomes and long-term expected values drive belief updating when measuring explicit reported beliefs. The authors then show that long-term model-based prediction errors are represented more in the dorsal striatum and ACC, while ventral striatum represents model-free prediction errors. This finding is novel in the sense that previous studies using different tasks of model-based and model-free behaviour (e.g. Daw et al. 2011) showed only ventral striatal effects of model-based and model-free prediction errors. I particularly like that there is no uncertainty about the model of the environment, in that the subjects are continuously reminded of the possible probability distributions of the urns, and that immediate reward is uncoupled from the long-term expected value.

We thank the reviewer for their appraisal of our task as being elegant and that our results are novel. Below we respond point-by-point to the remarks raised.

First, I think the abstract and introduction could be rewritten to appeal to a more general audience, be more specific and clear about the question, and use less jargon. For example:

The first line of the abstract suggests that the task might relate to some sort of trade-off between short and long-term benefits of a choice (e.g. temporal discounting), when actually the paper and task are about how a single outcome may convey both short-term gains/losses and long-term expected value, but there is no trade-off between a long-term and short-term option as such. It would be good to make this very clear.

Overall, the abstract and introduction are very abstract, should be re-written for more general audience, with less detail/jargon and more about the implications of the findings. For example, the authors use terms like 'model-free vs model-based evaluation', and 'the relative influence of information over reward', 'a priori full model knowledge', 'belief update' which are all quite jargony terms.

We have completely rewritten the introduction and specifically avoid the possibly confusing allusion to delay-discounting. We make it now clear that our task and study explains the influence of rewards on inference and beliefs, which are related to choices (see below for an additional analysis of the relationship between beliefs and choices), but constitute a separate entity. We believe that these changes have fundamentally strengthened both the clarity as well as the readability of the revised manuscript and thank the reviewer for the suggestions made.

The sentence "choose actions purely based on beliefs, which does not require prior experience and can sometimes lead to decision against pleasurable prior expectancies. "

is again very abstract: what do you mean by not requiring prior experience? Experience with what? And where does the belief arise from? And again "monkeys do not seem capable to process information independently." Independently of what? Furthermore, the claim that the 'ability to down-weight short-term outcomes.... contributing substantially to the evolutionary success of the human species' is rather a big one, not backed up with arguments, and not really necessary for the paper.

We agree with the reviewer and have rephrased the initial sentence. We furthermore have removed the admittedly speculative reference to the evolutionary success of humans, but simply point out that such a capability provides organisms with the possibility to develop complex, to a certain degree reward-independent, behaviours.

1. Within the blocks of 20 trials, marbles were drawn without replacement (i.e. the proportions were exactly the same for all blocks). Why was this the case? This makes the relationship of the short-term outcome and long-term expected value more complex, as the long-term expected value is the average of all the marbles that have not yet been drawn.

Is there any evidence that subjects used this information to adapt their strategy, i.e. pick the bad urn when they had 'used up' all the good options in the good urn? To test this, the authors could use the actual expected value (i.e. the mean of all the 'marbles' that are 'left in the urn' still), given the marbles that have already been observed in a block.

The reviewer is correct in that the structure of the events was always the same, but the outcome depended on the chosen urn. This was done to keep the informational content of every block identical, such that

every block contained the exact same number of congruent, incongruent and non-informative outcomes and ensures that trial numbers for the fMRI analysis diverge as little as possible. Only the valence of the payout was choice dependent, and a pre-determined incongruent event, e.g., would be positive if a *bad* urn is chosen (we made this clearer at the beginning of the Results section of the revised manuscript, see p. 4f). Thus, participants would have had to not only keep track of all the outcomes they have received in a block, but also if they had been drawn from a *good* or *bad* urn. Indeed, this leads to the possibility that if a participant would be capable of doing so, in the last few trials of a block the expected value of a *bad* urn may be higher than that of a *good* urn if more incongruent events / marbles were left. We firmly believe that none of our participants was capable of this, which requires to combine in parallel and retrospectively (during the first trials of a block, participants cannot know an urns long-term valence) combine outcome information, an urns true valence and the number of events that have been drawn, and the number of events that are still possible to draw. Furthermore, participants would have to believe that events are non-stochastic, which they were not told. To test this possibility, we included a regressor into the additionally performed analyses of choice behaviour described in response to the next point. We simplified the expected value (*EV*), because if a *good* or *bad* urn had the higher *EV* depended only on how many congruent compared to incongruent events were left in a block. If the effect of *EV* interacts with the time in a block, this could be seen as evidence for participants actively exploiting such a strategy, although this would require an astounding capacity given that solving the task itself was already quite challenging. In short, using logistic regression of choice of *good* urns, we do not find evidence that participants chose an urn in later stages of a block more in dependence of the expected value left for that urn (interaction *EV_left* x *Trial_no_in_block* $t_{23} = -0.38$, $p = 0.7$). There was, however, an overall effect of *EV_left* ($t_{23} = 3.35$, $p = 0.017$ corrected), yet this is most likely explained by the fact that congruent events had a stronger influence on choices. That is: if an event was congruent, it more likely led to subsequent choices of the better urn, as confirmed by the analysis of believe updating (Figure S1). We added these very informative additional analyses to the supplements of the revised manuscript (please see also the response to the following point).

2. My key concern is that the authors model the beliefs of the subjects, rather than the subjects' choices. Why is this? We have long known in psychology that introspection is certainly by no means an unbiased window on what drives our choices. Presumably the ultimate aim is to understand the subjects' behaviour, not their estimates of the current state (or at least not the explicit reported estimates, but rather the implicit estimates that drive choice). This makes the current paper very difficult to compare to any other papers that talk about model-based and model-free choice. Why not at least include an analysis of the choices as well as the explicit reported beliefs (and the associated neural effects?). How do the reported beliefs link to the explicit choices? Do you think that asking subjects to report explicit beliefs changes their choice behaviour?

We agree with the reviewer that predictions of behaviour are an essential goal in psychology, neuroscience, economics and related fields. Behaviour, however, and beliefs established via inference are without doubt in a reciprocal relationship and none can be fully understood without the ability to model the other one as well as their interrelationship. We therefore further fully agree that the addition of a dedicated analysis of participants' behaviour, and the relationship between belief and behaviour, provide a substantial extension to our study, and we included these in the discussion and supplemental material. We reproduce these results below. For results of neural correlates of choices during the decision phase of the task, please see our response to reviewer #3 point 8 below.

Furthermore, we conducted a control study in which we did not ask participants about their beliefs after every trial, but only intermittently (please see newly added section *Supplementary Discussion* with more details on the replication study including fMRI data from 18 novel participants). Using the same logistic regression model as below, we found that, just like in the original version of the task, choices were strongly influenced by beliefs: higher beliefs of the chosen urn were associated with more choices of the *good* urn ($t_{32} = 4.56$, $p < 0.0005$), and lower beliefs of the alternate urn to be *bad* ($t_{32} = -4.58$, $p < 0.0005$, see Figure R2 below for a display of choices depending on previously entered beliefs of chosen and unchosen urns). Unsurprisingly, however, the overall model fit was reduced (to $R^2 = 0.52$), because the belief regressor in many trials did not reflect the actual current trial's belief, but a belief prompted some trials before. Thus, while asking participants about their beliefs may induce slight changes in behaviour, these data clearly indicate strong similarity between choices when participants have to report their beliefs all the time, and when they do not have to do so (please see the *Supplemental Discussion* for more details about belief and

choice behaviour in the replication study). Therefore, we are confident that beliefs prompted in our task strongly relate to choice behaviour, and that inducing a prompt does not change behaviour systematically.

Figure R2: Data splits based on replication sample with only intermittent belief prompt. Splits are trichotomized based on participants beliefs entered (compare to figure below).

New analysis added to the supplements added as Figure S2:

Figure S2 | Relationship between beliefs and choices.

To investigate the relationship between participants' beliefs and their choices, we conducted a logistic-regression onto participants' choices of urns when one was good and the other bad. We

defined choices of good urns as 1 and of bad urns as 0. As regressors we included the valence of the previous outcome (-1 = loss of points, 0 = no change, 1 = gain of points), and the current trial number in the block (*trial no*) to account for general learning over time. Additionally, we included the relative expected value (*EV*) of the *good* or *bad* urn as if participants had knowledge of this, as well as the interaction *expected value* x *trial no*, to test if the participants might have counted events that had not been drawn. The relative *EV* depends on how many incongruent compared to congruent events are left. For example, had a participant counted all received events, and in the last trial only an incongruent marble was left, the better choice would be to select a *bad* and not a *good* urn. We first test the influence these factors have on participants choices in a logistic regression model, and then compare this model to another one that includes the belief of the chosen and the belief of the unchosen urn to be *good*. Furthermore, we plot raw quantile splits of the data, which show the effect of the included factors separately, that is, when variance by other factors is unaccounted for (b-e).

First, we find that inclusion of participant's beliefs dramatically increases model fit ($r^2 = 0.177$ without and $r^2 = 0.704$ with beliefs). Despite anticorrelation between the beliefs of chosen and unchosen urn (mean $r = -0.49 \pm 0.06$), we found that the strongest factor driving participant's choices in the task, is the belief of an urn to be *good* ($t_{23} = 9.25$, $p = 1.96 \times 10^{-8}$) over and above the effect of believing the other urn to be *bad* ($t_{23} = -6.31$, $p = 0.000016$). This confirms the relevance of beliefs for choice behaviour. Furthermore, because shared variance in regression analysis is attributed to the error term, this indicates that participants employed information about the alternative urn, and chose a good urn more often when the alternative was estimated to be poor, as is also confirmed by the raw data splits (b,c).

In the regression without beliefs, in accordance with the raw data plots (d,e), we found a significant positive effect of trial number on the proportion of *good* urn choices ($t_{23} = 5.97$, $p = 1.71 \times 10^{-5}$) and a negative effect of the valence of the previous outcome ($t_{23} = -4.36$, $p = 0.0009$). Both effects were fully explained by beliefs and no longer significant in the regression including participant's beliefs (both $p > 0.06$). This indicates that beliefs in the current task mediate effects of learning over blocks and previous trial outcome effects onto choices, further confirming the behavioural relevance of beliefs and inference in the current task.

There was a significant effect of the expected value left on choices of the better urn ($t_{23} = 3.35$, $p = 0.017$). This is most likely because the expected value depends on how many congruent events have been observed, and more congruent events lead to more choices of good urns as these are unaffected by bias. However, there seems to be an additional effect of this on choices that is over and above the effect on belief, possibly indicating that choices are even more biased than inference. There is no interaction between *EV* and trial number ($t_{23} = -0.38$, $p = 0.71$ uncorrected), indicating that participants did not count task events.

In (a) red horizontal line = median, box = quartile range, whiskers = range, error-bars in b-e = SE, p-values were corrected for the number of factors in the respective models by applying Bonferroni correction.

3. The authors contrast a model-based and model-free learning strategy, but also conflate this with a Bayesian and an RL framework, which is unnecessary and may be confusing, particularly for readers not very familiar with computational modelling. This choice has as a consequence that models are hard to compare because the Bayesian model is normative (no free parameters), whereas the RL model needs model-fitting (of the learning rate). Why do you not simply use a Bayesian 'model-free' model, ie a normative model that does not incorporate full knowledge of the environment (i.e. the pie charts), to contrast with the Bayesian model-based model?

Furthermore, in order to assess the relative and independent contribution of both model-based and model-free learning the authors use a multiple regression model that is partly circular: first, you fit a model using a learning rate, on the beliefs, and then you do a regression with prediction errors computed using the same learning rates.

We thank the author for pointing this out. In accordance with the comments of reviewer #3, we agree that the difference between the outcome- and expectancy-term that constitute an *RPE*, is of high importance. We assured that the effect on belief updating we found using the *RPE* from the RL model, can also be seen when the regression onto beliefs is repeated using only the outcome part of the *RPE*, which avoids the

problem of circularity as this is not fit to participants' behaviour. This, as well as the independent regression onto the fMRI signal using expectancy and outcome as separate regressors, indicates that *RPE* effects seen both on behavioural and neural level, are mainly caused by the actual outcome and not based on expectancy. We state this confirmatory result in the corresponding Methods subsection (Behavioural regression model, p. 17) and we also extended the part of the discussion of constituent components of *RPE* signals and their respective contributions to the results of the current study.

We have chosen an RL learner as a framework to compare model-free outcomes, because we feel that it is a well known, validated framework that is well suitable as a reference. We expect that many readers will have an immediate understanding of the implications of RL model and *RPE* signals, as well as their expectable neural correlates. Comparing two Bayesian models with each other would, even if one were very similar to an RL model, increase the complexity of the analyses as well as the required explanations considerably. We hope the reviewer agrees with us, that after avoiding the circularity, the comparison between the RL model and the Bayesian learner is a valid and relatively easily understandable basis for the analyses we perform.

Figure R3: We repeated the regression analysis of participants' belief updating using outcome valence instead of the *RPE* to strictly avoid circularity. Comparable to Figure 1D, valence exerted a positive effect onto belief update and participants updated in the direction of the experienced valence ($t_{23} = 3.76$, $p = 0.001$). This suggests that outcome drives the *RPE* effect on behaviour and we discuss this in more detail in the revised manuscript (see also Scholl et al., *J Neurosci*, 2015).

Results

Minor points

p_3 - 'likelihood ration of informative events remained constant over blocks - unclear what informative events refers to at this point.

Thanks, this has been clarified and we introduce the distinction between *informative* and *non-informative* events earlier in the revised manuscript.

make clear that in any given block, also two good or two bad urns could be present.

We put more emphasis on this in the revised manuscript. We added the following sentence to the task description presented in the results of the main manuscript on p. 5:

In one block both urns were good, and in another block, both were bad, which was always the third block in the experiment to avoid that participants generalised beliefs of one urn to the other one.

add how you determined that the excluded subject did not understand task instructions

We have added this to the manuscript. The participant always selected the left urn over multiple blocks, leading to completely random choices depending on an urns' position rather than identity (colour). The participant furthermore did not complete all questionnaires following the experiment and left pages blank.

p 11: nr of choices: that is the +/-? SD? range?

We added the correct measure (SE).

'n choices' - should this be filled out?

We corrected this to *number of choices*.

modelling: all payouts were scaled to the highest possible outcome per block - why?

We re-scaled outcomes to the highest possible outcome per block to reflect range adaptation effects that have repeatedly been demonstrated for diverse types of neurons (see for example Kobayashi et al., *J Neurosci*, 2010 or Padoa-Schioppa, *J Neurosci*, 2009). Additionally, due to the low learning rate, *RPE* and outcome are highly correlated, such that this apart from being an a priori justified scaling, does not affect results much. This is stated in the methods of the revised manuscript at p. 16.

Why did you not fit the choices, but rather the subjective beliefs? Choice and subjective belief are probably a readout of the internal belief state.

We did not fit choices directly because we intentionally decoupled choices and beliefs to some degree in the task because we encouraged participants to explore in order to win the bonus points between blocks (see results of regression analysis described above). Therefore, choices here are a less valid option for fitting: They do reflect beliefs, but at some points participants start to explore, not because their belief state changed, but because they aimed at acquiring knowledge in order to obtain the bonus points. The benefit of this is that it allowed us to keep the task more balanced. Had we encouraged participants *only* to exploit, this would lead to a much larger number of events depending on choices of the *good* urn (i.e., an incongruent event would mostly mean negative payouts and positive long-term inference). For this reason, we feel that fitting the model to the belief is the much better alternative than using choices and, additionally, the regression analysis confirms that outcomes are translated into choices via beliefs (see above).

rephrase: p 10: perceived reward/punishment  received change 'beliefs in the expected value' to 'the expected value'

Thank you, this has been corrected.

Reviewer #3 (Remarks to the Author):

The authors manuscript is a functional imaging study of an inference and learning task in humans, in which participants need to learn whether options are good or bad by integrating experiences in an appropriate way, and by avoiding inappropriate biases/use of information.

Overall, the manuscript is a very nice piece of work containing a very carefully analyzed data set, with a lot of thought going into understanding the cognitive processes and neural mechanisms underlying it. However, I believe that some of carefulness and the novelty is obstructed by aspects of the framing and the brevity of the interpretation/discussion section. In other words, by emphasizing specific novel parts of the results in the main manuscript and supplements more and focusing less on the already previously described aspects or interpretations (such as simple model-based vs model-free), I think, the manuscript could be significantly improved. Also, there is one particularly pertinent study missing in discussion or introduction (Scholl et al., 2015 JoN), which also looked at reward induced biases against long term, more optimal learning and highlighted frontal pole and dACC (here referred to as MCC). The study is sufficiently different from the current manuscript as to not affect the novelty, but is very relevant, while also suggesting some alternative explanations regarding some of the neural findings (see below). In short, if the authors emphasize more the idea of inferential processes and how such inferences interact with, other, lower order reinforcement learning and primary reward experiences, I think they have very exciting manuscript (to put it bluntly it explains the difference between “this is what happened just now” (Good vs Bad, rewarding content) and “this is what it means” (Urn identity/Casual structure, informational content)). The distinction between experience and inference would also make a clearer clinical link.

Other than including more results in the main manuscript (e.g. the trial-wise Bayesianness is excellent!) and extending the discussion I only had a couple of comments regarding some of the descriptions of the findings and minor comments about labels, wording etc. The fact my comments are long is a sign of my interest. It is my hope they will help the authors to make the novelty and importance of their findings clearer and will hopefully not be too much work, as I am not suggesting any large scale re-analysis.

We thank the reviewer very much for the positive assessment of our study. We have incorporated the suggested changes and extended the discussion of various aspects, included the trial-wise Bayesianness analysis into the main manuscript and changed the overall terminology to now more precisely describe the main aspects as the difference between reward experience and inference-based learning, that we subsume under the much broader term model-based learning. This is also reflected in the new title of the revised manuscript which is now „How reward experience biases inference“.

Major Comments:

1. It would help immensely with the readers comprehension if the task description was way earlier. I had a hard time understanding why the authors could dissociate the things they could from the text. If they had included a simple, common sense, description of the task early, this would have been easily avoided.

E.g.” We designed a simple binary decision task in which participants had to learn about two different urns in a blockwise fashion. However, as there was a more complex probability distribution of outcome magnitudes, both negative and positive, we were able to contrast the Informational content of a trial from its current rewarding nature. Simply put, a specific rewarding outcome could be more frequent in the bad urn leading both to a positive prediction error for the current outcome and an inferential (potentially bayesian) process of thinking it more likely it is the bad option. In other words, while participants need to process positive reward information they also need to assess the informational content of the reward, which might lead them to make a negative conclusion.” Also Related to clarity, maybe explain distinction between informative and non-informative events earlier. Maybe even a figure/Illustration in the main manuscript about inference, specific events and learning might be nice. Similarly, some of the points below might make it necessary to move some of the analysis into main figures, such as Figure S5.

We thank the reviewer for this very helpful advice. We have included a similar easily understandable task description in the introduction, and moved the explanation of the difference between informative and non-informative events further at the beginning of the text. We feel that this has greatly increased the accessibility of the manuscript, task and results.

2. As mentioned above, I think Scholl et al. 2015 in JoN is a very relevant paper and should be discussed, as it showed that in a learning task when receiving irrelevant reward information, the effect-size of that reward signal at outcome in frontal pole and in their case dACC/MCC was predictive of how unbiased participants were. The authors of that study however interpret their effect in terms over the signal being necessary in order to overcome, suppress or compensate for such biases, rather than integrate it (see : “integration of model-free signals within regions coding Bayesian model update.”). Furthermore, in the Scholl study the authors regress against the outcome not a RPE. Seeing that RPE’s are strongly driven by outcome signals, how much of the irrelevant/model-free reward signal preventing biased behaviour is driven by the outcome? I do not expect the authors to rerun all their analysis just with outcome, rather than RPE, particularly as they show some of the regions to have expectation suppression, but was curious what their sense was, in case they looked at the difference at some point. Also, even if it was mostly about contextualizing reward experiences rather than a “real” RPE signal for learning, the process should get easier with repetition, giving a RPE like effect of reduced signal with repetition, without actually representing an RPE-like learning mechanism. In short, have a proper discussion of Scholl et al., and the implications (i.e. suppressing irrelevant information) and maybe briefly mention RPE vs pure outcome processing.

We thank the reviewer for the additional reference to the study of Scholl et al., which we added to the discussion. We completely agree that the functional activation by model-free reward / outcomes associated with reduced influence of RPE on behaviour, most likely represents suppression of this information, and rephrased „integration“ to „representation“ in the results, which does not imply how this activation functionally relates to learning.

We have not explicitly separated the across participants regression to differentiate between RPE and outcome signals. However, due to the low learning rate of the model, the RPE regressor here certainly reflects mostly the outcome and we interpret the correlation across participants, as well as RPE main effects, as outcome driven. We have made this clearer in the revised manuscript.

This section has been added to the discussion on pp. 11f:

Such a pattern of activity which increases both with inferential updating and reward signals related to the outcome, are well compatible with suppression of model-free signals, which seems to enable some participants to overcome biases in belief formation better than others. These results fit nicely to a recent study in which participants could employ information conveyed either via real or hypothetical outcomes, which in itself was irrelevant for learning (Scholl et al., 2015). However, participants’ decisions were biased by real outcomes, yet activity in lateral FPC and pMFC counteracted this bias. The exact mechanism, how competition of short-term and long-term information could be solved in overlapping or neighbouring neuronal ensembles in the mediodorsal caudate and the FPC remains to be elucidated in future studies integrating biophysical models of decision making with knowledge about cortico-striatal circuits (Wei et al., 2016) as well as the transfer of information between different striatal sub-compartments, e.g. via spiralling connections through the dopaminergic midbrain (Haber et al., 2000).

3. Putting expectation and outcome into a regression is very nice and beyond the level of proof most other learning papers offer! However, why the expectation is based on the Bayesian learner needs to be discussed more. See: “the model-free single-trial RPE most likely based on expected value estimated from model-based beliefs”.

If I am not mistaken this makes what they call the model-free RPE not actually model free, as the expectation is based on the optimal Bayesian belief structure. This has implication for their framing. More specifically, it implies that although vSTR does process primary reward information, it is nonetheless controlled/modulated by expectation that includes higher order beliefs. It also suggests that even though the region does not represent higher order reward signals such as odds of being in the good urn, it does modulate its expectation of something good happening by whether the option is seen to be the good or bad one according to the inferential process, making it sensitive to more “cognitive” inferential modulation. If the authors discuss this bit more their nice differentiation between different expectation suppressions in Figure S2 might be nice in the main text. It could also require a discussion of why DKL regions don’t hold previous belief but only update. In this context it might be nice to cite Boorman 2016 in Neuron, as they found IOFC signals of updates but not representation.

Yes, we agree that *regions*, specifically the ventral striatum, cannot be interpreted as processing *only* model free outcomes, although the regressor itself is model free. We made this clearer in the revised manuscript and incorporated the reviewers suggestions as well as the very informative and related study by Boorman et al. into the discussion. The corresponding sections on p. 10 is as follows:

Consistent with the idea of information transfer to the dopaminergic system (Doll et al., 2009), we found that the ventral striatum integrates beliefs and payout. This indicates that the ventral striatum does not reflect a learning mechanism that is strictly model-free, i.e., based on reward experience, but incorporates beliefs obtained by inference (Fig. S3GH), compatible with previous studies (Daw et al., 2011). Relatedly, we found that the dorsal striatum only weakly represented current beliefs at outcome (Fig. S3c,d) while reflecting updating, indicating that different brain regions may process values and update terms (Boorman et al., 2016).

4. Related to point 2, a lot of Model-based vs model-free papers ignore the fact that in order to be model-based a person needs to potentially suppress the reward experience (e.g. Daw's famous two step task). In other words, rather than just talking about model-based use of transition probabilities, I think it is important to highlight in this manuscript and traditional model-based vs free tasks the fact that the cognitive phenomenon of suppression of reward information also exists. I think this might be important for understanding particularly the effects in this study of activity in frontal pole and other places that is both very Bayesian and activity that is strong when reward experiences are strong, as this is when interference needs to be avoided. Related to this, while the authors of this manuscript say DKL + RPE in dorsal striatum makes participants unbiased because of "integration" into learning, it could be precisely because of the opposite of not including it into a value estimate, but keeping the interference lower in the value learning system. Related to this there might be a title that includes the possibility of such a mechanism explaining the co-occurrence of long term/inference based value and short-term/reward experience based representations. [As a side thought, it ultimately might be interesting to measure physiological inhibition effects with e.g. MRS and see whether inhibition in target area correlates with less interference].

We have put more emphasis on this interpretation of the findings, and we fully agree that suppression of reward and reward-biases is the most likely functional explanation for the covariation between regions involved in belief updating as well as model-free signal processing. Indeed, we did not intend integration to be meant exclusively as actually employing information in a positive way, but more in a sense that it reflects a process that integrates both, inference-based information and model-free reward experience, into an output, leaving open the interrelation between both. However, we realise that this terminology can be misleading, and we have put more emphasis on the importance of suppression of irrelevant reward processing.

Additionally, we share the reviewers interest in comparing physiological measures of inhibitory compared to excitatory signalling to test the hypothesis that these regions actually facilitate suppression of reward biases more directly.

5. I think the authors could make a bigger deal about Figure S5, as they can only look at it because they have the nice trial-wise quantitative rating after every trial. It is a very interesting result showing that if participants in a trial, manage to quantitatively update in a more Bayesian way, the dACC/MCC is engaged together with a network of other regions, such as dmPFC/medial Frontal Pole, posterior cingulate etc.. Many of those regions are interesting for people looking at dynamic decision making and learning and will be interested in this finding.

We agree with the reviewers suggestion and have moved the analysis of single-trial Bayesian updating into the main manuscript (presented in Figure 4 of the revised manuscript).

6. While many papers have talked about model-based and free and Scholl et al, talked about reward related biases and overcoming them, the striatal results are very carefully done and novel. The fact that the striatum isn't a unitary structure is very cool, but I think it might be nice if the authors spent a bit more time to relate their finding a little bit more to other ideas of dissociation between dorsal and ventral e.g. Trevor Robbins' work on compulsivity and habits and the ventral to dorsal transition. It wasn't clear to me how to square Robbins results with the current finding, but it is a well-known theory related to learning and it might be good if the authors said how they might relate.

We have added this important topic to the discussion. However, we feel that our results are difficult to compare to the concept of habit formation associated with a shift of behavioural control from ventral to dorsal striatum, because the definition of a habit does not apply to either form of learning. It may, however, be possible that the dorsal striatal activity can be interpreted as an (early) formation of a habit, in that participants try to *recode* outcomes to reflect purely the informational content and abstract the reward properties. It would certainly be an interesting follow-up to measure changes in differential striatal responses following prolonged training of one distinct probability distribution. We have added the following to the discussion:

A functional dissociation between ventral and dorsal striatum has been associated with formation of habits, i.e., behaviour resilient to outcome devaluation (Everitt & Robbins, 2005), and the dorsal striatum is supposed to control habitual responses. In this framework, the activation in dorsal striatum we see may reflect formation of a habitual response that could over time replace active inference. It would be informative to test this hypothesis by exposing participants over longer time periods to the same event distributions, such that events could directly be associated with belief changes, possibly by-passing active inference.

7. I realize this might be more of a personal grudge of mine, but I find the term “model-based” terribly generic to the point of being meaningless. Even worse, it often obscures the novelty of a finding and does not highlight the functional significance enough. For example here, model-based inference relies on proper contextualization of a rewarding outcome, suppressing of the immediate reinforcing properties of reward and a statistical inference of an observation, while other “model-based” tasks such as Doll et al 2016 Nat Neuro, is called model-based but there model-based means generalization between two stimuli which both lead to the same outcome.

We agree with the reviewer that the overall term is being used very broadly. As mentioned above, we have changed the framing more to a contrast of inference and model-free experience. We discuss the different possibilities and implications in a separate section of the discussion, and adopted the reviewers example.

8. Were there any relevant/interesting decision-related signals in the task? I do believe the authors dissociated the outcome from decision phase temporally, but do not mention whether there were any interesting effects related to those analyses. E.g. was there an effect of last trial update on the decision phase? Or alternatively, was there a larger FPole signal when making decisions and having to go against a last trial rewarding, but ultimately/statistically bad urn?

Our design is certainly suboptimal to study stimulus related activity and the task aimed at investigating outcomes and belief updating / learning. The delay between response and feedback was 1 s and participants could choose an urn whenever they were ready (median RT = 673 ms \pm 37 SE) and thus decision related activity is difficult to fully separate from outcome related activity. Furthermore, because participants had unlimited time to enter their beliefs following each feedback, decisions may have in many cases been already formed before the actual trial onset which additionally was fully predictable apart from the side on which an urn was presented (which was randomised).

Nonetheless, we ran a GLM including regressors at stimulus onset that coded for:

- the belief difference between both urns
- if an urn was repeatedly chosen or if the response was switched
- if the decision was against the higher belief (i.e., if the decision was exploratory)
- as well as unspecific factors similar to the main analysis (e.g., responses, belief prompts from the previous trials etc.), always additionally modelling the effects of D_{KL} and RPE at feedback time to reduce possible overlap with decision related activity. Note that the outcome related effects were not qualitatively altered by including the factors noted above into the GLM.

Briefly, we find effects of the difference in beliefs between both urns in pMFC (Fig. R4, top row), akin to an inverse value difference signal (Rushworth et al., *Curr Opin Neurobiol*, 2012). Furthermore we see additional increased BOLD signals in pMFC, dIPFC and IPS when participants switch selection of urns, compatible with a cognitive control network (Fig. R4. second from last row).

Additionally, we find negative effects of the chosen belief in very similar regions as seen for D_{KL} (Fig. R4 bottom row) at outcome phase. Depending on threshold, such an effect was also seen in the DS, albeit weaker than in the VS and especially the cortical regions.

We did not find increased FPC signals when participants chose an urn following a negative payout again, possibly because the actual decision which urn to chose may have been made during the belief evaluation of the previous trial and the signals we see are rather anticipatory. We do find that the negative covariation with chosen belief extends into FPC, which appears related to the effect the reviewer suggests: FPC increases in activity when participants actively chose a low value / *bad* urn.

Given that our design is not well suited to distinguish decision and outcome phase, that the manuscript is already at the maximum length limitation and that other reviewers additionally suggested further analyses, we did not include these results into the revised manuscript. If, however, the reviewer suggests that we should report decision related activity, we would be happy to include a short summary thereof in the supplemental material.

Figure R4: Results of an analysis of decision related neural signals. All decision related effects are cluster corrected at $p < 0.05$.

Minor Comments:

A) Page5 “assuming gradual change (Fig3D)” gradients are very hard to prove, particularly in fMRI, as gradients could be an artifact of movement combined with discrete clusters. I don’t think a gradient is particularly important for their argument, but I would just recommend a bit of caution about gradients.

We agree with the reviewer and added a sentence cautioning the readers about over-interpreting the gradient results we present at p. 7. The following has been added:

Although it is difficult to fully rule out artefacts as the cause of gradual effects in fMRI, ... (this finding is well compatible with the view of information integration from ventral to dorsal striatum via spiralling reciprocal ascending midbrain projections).

B) The figure legends in Figure S1 seem off/incorrect. E.g. in C the p values are tricky to read. One says $p < 10$ with the -3 below the 10.

We apologise for this inadvertency, this has been corrected in the revised manuscript.

C) More labeling of figures might be nice. For example labeling the colours in figure 3 in the figure itself might be nice.

We appreciate this suggestion and have added more labels to the figures which we feel considerably improved accessibility of the graphics.

D) When the authors say “Understand how both learning mechanisms are implemented in the brain.” The sentence is slightly confusing as it is hard to reconstruct what both refers to.

This sentence is no longer included in the revised manuscript.

E) The statement “Bayesian solution to the problem which was easily accessible to participants” is quite a strong claim. It is true that the statistical information was always present but explicit Bayesian solutions are rarely trivial. Maybe rephrase and say, “the information for Bayesian statistical inference was readily accessible and participants seemed to be guided by Bayesian like updating”

We agree and this was an imprecise wording by us. We meant that the overall problem and task was easily accessible as all participants were able to solve it successfully without much training. We rephrased this sentence to:

The two-urn task provided an ideal Bayesian solution to the inference problem and was easily accessible by the participants.

F) Is former later wrong way around in the abstract or are the authors saying short term in dmSTR and FPole and long term in vSTR?

You are right, this had been the wrong way and has been changed in the revised manuscript.

G) The authors say “Which model applies to current context” which suggests model switching. It might be nice if the authors elaborated a little bit more, as they might also have meant which urn/ statistical environment applies instead of switching “models”.

This part has been rewritten and clarified in the revised manuscript. We were referring to the statistical environment in the task.

H) “Negative covariation between the behavioural influence of model-free prediction and the steepness of the RPE gradient” could be maybe said with viewer double negations. Took me a while to understand.

We have clarified this sentence.

I) I Didn’t find it needed the IGT discussion very much, as it is a rather outdated experiment. I however understand if the authors want to keep it.

We tried to abbreviate this part of the discussion. However, we were repeatedly asked about similarities when presenting the data at conferences, so feel that readers will benefit from a brief discussion of the relationship between this task and the IGT.

J) It might be better to discuss the frontal pole subregions and general overlap in an anatomical region qualitatively rather than focus on the very few voxels that survive formal overlap analysis, but again, I defer to the authors preference on this.

We agree that the formalized overlap analysis is not always the best tool to illustrate conjunction effects. However, we feel that most of the overlap is large enough to be interpretable and otherwise we additionally present the main effects for the analysis, such that the overlap should be accessible for the readers.

To summarize, I congratulate the authors on a very nice study. With some reframing of the findings and inclusion of some more of the already done analysis in the main manuscript and some intuitive illustrations (as described in my major points below), I think this study can be very interesting for cognitive neuroscientists as well as a wider ranging community including theoretician and clinical researchers.

Again, we thank the reviewer very much for the constructive and very helpful comments, suggestions and questions. We feel that the manuscript has improved considerably with regard to clarity and interpretation of the results.

Reviewer #4

We very much appreciate the thorough summary, review and criticism of our work by the reviewer. In short: To satisfy the criticism that a sparser explanation of our results and data would be that participants did not actually learn values / form beliefs, but made decisions mainly based on spatial representations of the belief bars which we provided, we conducted a control fMRI analysis. We changed the task such that a) belief bars did not maintain their previous positions (thus, participants had to integrate beliefs) and b) beliefs were not prompted following every trial (participants needed to update representations that were not prompted). Briefly, the fMRI results demonstrate significant effects in all regions identified in the original analysis. We furthermore conducted additional analyses to investigate the role of how beliefs drive choices more clearly, and that confirm that participants successfully learned the task as intended: they generated beliefs about the underlying long-term properties of the urns, by exploring not only *good*, but also *bad* urns, ultimately exploiting both within block choice possibilities and bonus points for correct identification of the urns.

Below, please allow us to correct some minor misunderstandings of the task, and respond to the points raised in detail.

We feel that the critique of the reviewer and the incorporation of replication samples on behavioural and imaging sides considerably strengthen the manuscript.

SUMMARY

Long-term versus immediate outcome information

(...) Dissociating immediate and long-term outcomes created the possibility of looking at congruent versus incongruent cases to better assess the differential contributions of each type of consequence. On some trials, the immediate negative or positive feedback sum was not presented on the pie-chart and therefore these trials were uninformative of the long-term association.

Please allow us to correct this misunderstanding. All feedbacks were always present in the pie charts, but non-informative events were equally likely under both distributions. Therefore, the difference between informative and non-informative events is only their likelihood-ratio conditional on the urn's valence. We have made this, as well as the task description in general, much clearer in the revised manuscript.

Earning/losing points

(...) Confusingly, though they were encouraged to explore both urns to be able to earn the 10 bonus points, they were also advised to 'maximize their earnings by prioritizing good urns'.

We think it is very important to clarify why we chose these specific details for our task and why we feel that this is an appropriate design. We instructed participants that their goal was to find out if each urn was *good* or *bad*, for which they would be rewarded. They were furthermore correctly informed that choosing a *good* urn more frequently than a *bad* urn additionally increases their payouts. Thus, a plausible strategy for the task would be to firstly try to identify an urn as *good* and gain confidence in this estimate (ensuring the bonus money). Thereafter, one might figure out the valence of the alternative urn. If one gains sufficient confidence in this to answer the question between blocks, the ideal strategy is to exploit the *good* urn.

The reason to choose such a design is that we needed to avoid strong imbalances in the task. If participants were instructed to purely exploit *good* urns, they would only very rarely form beliefs about negative long-term consequences and the analyses would thus be strongly skewed. This is made clearer in the revised manuscript and we hope the reviewer agrees with us that this design serves the purposes of the study well.

Major Criticisms

1) This is an incredibly complicated task, with numerous components and important details, and though summarizing the task clearly and in a manner that is easily accessible is a challenge, it is essential for readers to be able to critically review the study and for the study to be replicated. I devoted considerable time to piecing the study together and have struggled to faithfully summarize all components here.

Overall, I would recommend a high-level overview of the task, followed by sections, separated by subheadings, emphasizing and clarifying the important features of the task and its implementation. Try to avoid 'vice-versa' and spell out the different conditions. Use consistent naming throughout as well to simplify.

We very much appreciate the reviewer's suggestions and have spent considerable effort in unifying the terminology and presenting the task, and the idea behind it, in a more accessible way to the readers. We

included a high-level overview, explicating the idea behind the task in the introduction. We then begin the results section with a detailed explanation of the task. We kindly refer the reviewer to pp. 4f of the revised manuscript.

'Participants were informed that they could maximize their earnings by prioritizing good urns.... the exact expected value per choice per urn was +1.5 and -1.5 points in good and bad urns, respectively.'

This is difficult to grasp on first pass and should be clarified further, with a specific example perhaps and by adding a label to specify this in the figure. Further, consistently naming conditions or outcomes could simplify. Elsewhere they refer to short-term or immediate feedback as 'payouts'. Here 'payout in points' per choice might be easier than 'exact expected value per choice'.

We have clarified what was meant by expected value per choice (the overall long-term expected payout that defines an urns valence), and clearly differentiate it from the immediate payout.

'Prior to scanning, all participants performed a training block of 20 trials after which they could discuss possible questions with the experimenter. During the task, the type of event (informative, non-informative, congruent or incongruent) was predetermined to keep the amount of information constant between all blocks and subjects. Thus, each trial was observed with a frequency matching its exact probability in a block indicated in the pie charts. However, the payout depended on participants' choices, such that a congruent event was associated with a positive payout if a good urn was chosen, and vice versa if a bad one was chosen'

Is this predetermined proportion of event type applicable only to the block of trials in the practice session? The paragraphing suggests it but within the paragraph the reference to 'all blocks' would suggest that this is referring to the experimental blocks perhaps as well as the practice block. Given the complexity of this experiment, avoiding anything, such as above, that adds ambiguity will be very important.

In addition to the proportions of event types, was the trial order pre-determined? Order will have an impact on how distinct RL and Bayesian Learner models develop over the block. Because urn choice was participant dependent, fully controlled trial orders, event-types could not be achieved.

We have clarified the wording here. Yes, all proportions and trial orders are predetermined as far as possible given that participants choices cannot be foreseen. Thus, for every participant the task consisted of the exact same sequence of information provided (congruent - non informative - incongruent etc), but the valence was dependent on the choice made per trial (congruent was positive if a *good* urn was chosen, negative otherwise). We provide an example for this in the methods (p. 5) that we reproduce here for the reviewers convenience.

The exact order of events was predetermined with regard to the information conveyed (congruent, incongruent, non-informative), but the valence depended on which urn a participant chose (e.g. a congruent event would be positive if a good urn was chosen on that trial).

2) *In the design of the task and in the introduction to the topic, the authors confound learning and deciding on the basis of immediate versus long-term consequences. Even the real-world examples provided in the introduction confuse these concepts. Rather than presenting situations in which problem behaviour results from difficulty simultaneously learning from short- and long-term consequences, both examples provide evidence of deficiencies in prioritizing long-term consequences over short-term ones in making decisions that guide behaviour. The authors themselves state plainly that people 'know' both the short-term and long-term consequences of their behaviour but are unable to resist behaviour that has positive immediate but negative long-term consequences or conversely in enduring short-term losses to achieve long-term gains. In effect, they describe scenarios that have generally been used to explore the delayed discounting effect in the study of decision making (Ainslie, 1975). Unfortunately, these confounds carry through from the introduction to the experimental task.*

We have changed the examples and framing of the study. We agree that our task does not measure the type of decisions which delay discounting paradigms aim at, but measures formation of beliefs *via* model-free learning, inference, or their interplay. We made this clearer and now use a much more consistent terminology. Overall, learning and decision making can in principle not be completely separated as decision making relies on values which have to be established via learning – either by immediate experience of inference – and learning can only be measured via decisions. We respond to the more detailed critique below.

3) In the current task, as implemented, participants are asked to make decisions on the selected urn's likelihood of being associated with a long-term positive or negative consequence, referring to a pie-chart that explicitly presents these conditional probabilities and was visible throughout the experiment. Correctly performing these decisions requires no learning at all. In contrast, participants had to resist being influenced by the valence or magnitude of the immediate/short-term feedback in making their judgment of the long-term consequence associated with the selected urn on each trial. Participants could perform this task optimally and generate the correct long-term consequence without learning anything, simply by making decisions with reference to provided conditional probabilities.

On each trial, participants indicated their judgment about the long-term consequence of the urn that they were evaluating, by moving a marker up or down in increments of 10 along a 0-100 continuum, with greater than 50 being 'good' and lower than 50 being 'bad'. At the beginning of each block, the marker for the blue and the marker for the yellow urns were set to 50. After each trial, participants moved the marker associated with the yellow or blue urn, depending on which urn had been selected for exploration, to reflect their decision regarding the probability that the urn was associated with a positive versus a negative consequence. Of importance, the position of the marker for each the blue and the yellow urn was conserved across trials. In this way, even to answer the bonus question correctly at the end of each block, no true trial-by-trial learning or creation of a long-term model was required regarding yellow or blue urn's 'good' or 'bad' status. The marker provided an explicit visual representation of the culmination of all judgments good or bad for each urn. To answer the bonus correctly, participants would have only to recall whether the marker was above or below 50 on the final trial to accurately perform the bonus question.

Although some learning/internalization of a long-term model was not precluded, as the task was designed, learning this long-term outcome was not required to accurately perform. Consequently, the authors' contention that they were directly studying belief formation itself was not warranted and the interpretation of the fMRI data is compromised by this ambiguity. In the larger literatures, DS, IPS, and dlPFC, which were associated with Bayesian learning/belief formation model in fMRI analyses, are previously extensively linked to decision making, particularly when influences on selections are ambiguous. The Bayesian Learner model could equally have been called the Bayesian Decision model, revealing higher activity in brain regions with more optimal decision strategies that are freer from bias from irrelevant feedback. All analyses related to the Bayesian decision/learning parameters can be re-interpreted in this way. Of brain regions that preferentially correlate with the Bayesian model, only the DS is also frequently implicated in learning. Increasingly it is being suggested that DS actually mediates decisions but paradigms often confound learning and decision processes (Hiebert et al., 2014, Neuroimage; Atallah et al., 2007, Nature Neuroscience).

We thank the reviewer for this very thoughtful and critical comment. We agree that especially leaving the marker in place after each trial may facilitate a strategy in which participants do not actually internalise the belief about an urn's long-term valence. We feel that there are clear indications that speak against this alternative explanation, for example the demonstration of covariation between belief and ventral striatal activity speaks for an explicit representation of this in a value-related manner. Furthermore, we see clear neural correlates of chosen beliefs at the decision time (please see response to comments of reviewer #3 comment 8 above). Additionally, follow-up surveys of the participants provided no qualitative evidence for such a strategy and all participants indicated that they tried to learn long-term valences (not remember the bar position at the end). However, we admit that our design could not clearly rule this possibility out.

To this end, we conducted a behavioural and fMRI control study including 16 and 18 novel participants each of which we report the results of the fMRI study in the revised supplements (purely behavioural results were similar). We introduced two important changes to the task: firstly, the belief bar marker always returned to 0.5 (indifference point) when a participant was prompted to enter her/his current belief. If participants still learn the task, this requires formation of beliefs in memory. Second, we prompted beliefs only on every third trial. Thus, if participants only decide to update a bar in one direction, one would not assume to find neural correlates of belief formation when participants know they will not be prompted about their belief on a given trial. In other words, if the neural correlates of belief updating (the D_{KL} regressor) remains similar between the original study and the replication, this speaks for a representation of learning rather than purely decision-related activity, because participants know they will not have to report a belief on a trial at all. The reason why we did not design the task in this fashion in the first place, is that it does not allow to assess belief formation directly in two thirds of the trials. This would preclude to

test the influence of incongruent events on belief formation without acquiring considerably more data (because the measure used for fitting would be patchy).

We added the results of this control analysis to the supplements together with a discussion of the reviewer's concern. We also reproduce these new supplements here for the reviewers convenience. In short, we reproduced all relevant effects for belief formation in crucial brain regions identified in the initial study including DS, aMCC, dlPFC, IPS, and FPC. These effects were mainly independent of whether participants expected to be prompted about their beliefs or not. Finally, participants successfully identified the urns' valences very similar to the initial study, providing strong support for an interpretation of results of the original study based on belief formation and updating.

Supplementary Discussion

Replication study - Behaviour

A potential confound of the task design is that the belief bar marker that participants moved to indicate their beliefs following each trial, remained in place. Thus, it would be possible to solve the task without forming actual beliefs about long-term valences of the urns, as it would be sufficient to decide in which direction to move the marker and remember its final position to solve the estimation of urns between blocks correctly. This appears especially important, because the dorsal striatum has repeatedly been implicated in decision formation processes, rather than learning^{10,11}. Although follow-up surveys did not indicate that participants used the (spatial) position of the bar as the basis for their choices, we conducted a replication study to rule out this possibility and ensure that the neural correlates of belief updating are truly related to inferential learning.

To this end, we introduced the following changes to the task. Firstly, participants were no longer prompted about their beliefs after every trial, yet only after every third trial. They could thus predict whether they would be asked to enter their beliefs, or not. This tests if prompting beliefs changed the neural effects we found. Secondly, the belief prompt was always set back to 0.5, the point of indifference. Thus, participants had to form and maintain beliefs about good and bad urns and could not rely on the spatial information provided by the prompt. This therefore excludes that the task could be solved without an internal representation of long-term valence of the urns.

We measured an additional sample of 21 participants for fMRI analyses, out of which 18 (mean age 23, 13 female) finished the recording session (3 left after the task was completed, but before the structural image could be acquired due to tiredness or strangury). All participants were informed about possible risks of the measurements prior to participation, and all procedures were carried out in accordance with the declaration of Helsinki. The study protocol was approved by the ethics committee of the medical faculty of the Otto-von-Guericke University, Magdeburg (Germany).

Participants earned on average 118 ± 9 (SE) points, and chose the *good* urn when one was *good* and the other one was *bad* (GB blocks) on 132 ± 3 and the *bad* one on 108 ± 3 trials ($t_{17} = 3.65$, $p = 0.0002$ for difference). Thus, they chose the better urn on slightly fewer trials compared to the original study (mean 21 trials, $t_{40} = -3.3$, $p = 0.002$), yet still successfully more often than the *bad* one.

As in the original study, participants successfully formed beliefs about *good* and *bad* urns (Fig. S6a). At the last prompted trial of *good* urns in GB blocks, they entered a belief of 0.667 ± 0.021 (t-test against indifference (0.5) $t_{17} = 8.84$, $p < 10^{-6}$) and on *bad* ones 0.419 ± 0.019 ($t_{17} = -4.36$, $p < 0.0005$). In blocks where both urns were *good*, this was 0.58 ± 0.028 ($t_{17} = 3.11$, $p = 0.0064$) and when both urns were *bad* this was 0.394 ± 0.027 ($t_{17} = -4.04$, $p = 0.00086$). Thus, participants established robust beliefs about *good* and *bad* urns even when they could not rely on the previous position of the belief marker.

Again, similar to the original study, participants successfully identified the urns' long-term valence between blocks to obtain the bonus points. For good urns, they were correct 10.9 ± 0.3 , wrong 0.6 ± 0.2 and responded that they did not know 0.4 ± 0.17 times (Figure S6a). For bad urns, they were correct 11.1 ± 0.2 , wrong 0.3 ± 0.14 and responded that they did not know 0.6 ± 0.19 times. None of these numbers was significantly different from the original study (all p for comparisons > 0.2 , uncorrected). This indicates that while participants were slower to establish beliefs, indicated by slightly less choices of the *good* urns, they still formed robust beliefs about the urns' long-term valences, likely because of the increased difficulty when the belief has to be remembered during trials.

Replication study – fMRI

All data for the replication was collected on a 3T Siemens Skyra scanner and preprocessing was identical to the original study, with the exception that no field map correction was applied. We again used an isotropic resolution of 3 mm, TR = 2s, TE = 30 ms, and a flip angle of 80° for the functional recording. The mean number of

acquired volumes was 1590 (range 1388 to 1999), mean task duration 53 minutes, comparable to the original study. Settings for the T1 MPRAGE were TR = 2320 ms, TE = 2.96 ms, TI = 1200 ms, otherwise identical to the original protocol.

As we could not fit the RL model to the participant's sequence of beliefs, because these were not available on a trialwise basis, we used the best fitting learning rate from the original study (0.06) to derive the *RPE* regressor. We used the same Bayesian model as in the original study, based on the belief participants entered on every third trial. The GLM was otherwise set up identically, but included an additional regressor that coded if a belief would be prompted after the feedback period, as well as the interaction between this regressor and D_{KL} , RPE_t , and ΔB_t , respectively. With this we test the hypothesis that beliefs are only formed, when participants know they should be entered afterwards. If the interaction is not significant, this indicates that belief updates and their neuronal correlates were not confined to prompt trials.

First, we replicated the striatal dissociation found in the original study (Figure S6b). We also replicated all major main effects seen for D_{KL} and these overlapped with the D_{KL} effects in the original study even when a whole brain cluster correction and a p threshold < 0.001 was applied. We found significant effects in the left (Fig. S6c, peak $z = 3.72$, $p = 0.0001$, $x = -9$, $y = 10$, $z = 10$ mm peak) and right dorsal striatum (peak $z = 4.06$, $p = 2.5 \times 10^{-5}$, $x = 7$, $y = 15$, $z = 2$ mm), left (peak $z = 4.07$, $p = 2.4 \times 10^{-5}$, $x = -35$, $y = -50$, $z = 43$ mm) IPS, left dlPFC (peak $z = 4.9$, $p = 9.6 \times 10^{-7}$, $x = -35$, $y = 11$, $z = 33$ mm), pMFC (peak $z = 4.6$, $p = 4.2 \times 10^{-6}$, $x = -4$, $y = 25$, $z = 43$ mm), and left FPC (peak $z = 3.61$, $p = 0.0003$, $x = -31$, $y = 52$, $z = 17$ mm).

The effects for *RPE* were somewhat reduced, possibly because of the worse fit possibility of the RL model. We still found overlapping significant effects in the left (Fig. S6d, peak $z = 2.50$, $p = 0.0124$, $x = -14$, $y = 17$, $z = -6$ mm) and right ventral striatum (peak $z = 2.77$, $p = 0.0056$, $x = 13$, $y = 19$, $z = -8$ mm). Furthermore, we found significant effects overlapping with the original cluster spanning vmPFC (peak $z = 2.26$, $p = 0.024$, $x = 20$, $y = 45$, $z = -3$ mm) and FPC (peak $z = 2.67$, $p = 0.0076$, $x = -2$, $y = 65$, $z = 6$ mm).

The only significant interaction effect between belief prompting and D_{KL} or *RPE* was seen in the left IPS (peak $z = 3.3$, $p = 0.00097$, $x = -42$, $y = -46$, $z = 42$ mm), indicating that D_{KL} was more strongly reflected here when participants knew that they would be prompted to enter their current belief estimate afterwards. At weak thresholds, interactions also emerged for the left dlPFC (peak $z = 3.06$, $p = 9.6 \times 10^{-7}$, $x = -45$, $y = 21$, $z = 31$ mm), pMFC (peak $z = 2.9$, $p = 4.2 \times 10^{-6}$, $x = 3$, $y = 22$, $z = 53$ mm), but not the dorsal striatum or the FPC. In sum, these data indicate that participants updated beliefs mostly independent of whether they would be prompted to enter them in a trial, as is appropriate to solve the task. The replication furthermore suggests that in both tasks, participants formed internal representations of belief states which were updated via inference derived from the payouts and knowledge of the event distributions.

Figure S6 | Results of the replication study.

We replicated the main results of the study in a second sample of 18 participants. In this version of the task, participants knew beforehand that they would be prompted about their beliefs only every third trial and additionally, the belief marker was always reset to the indifference point. Although this task was more difficult to solve, participants validly identified if urns were good or bad in the long-term (a). We replicated the striatal dissociation (b, blue = D_{KL} , red = RPE), as well as all D_{KL} main effects (c, original = blue, replication = green). RPE effects were weaker, but still present at lower thresholds, possibly because the RL model could not be fit to participants without prompting their beliefs.

Replication effects are displayed at uncorrected $p < 0.001$ for D_{KL} , and $p < 0.05$ for RPE effects.

4) ΔB_t , change in signed belief, was calculated and used in behavioural analyses but then is no longer referenced in the fMRI analyses. Were the analyses with ΔB_t similar to those with D_{KL} as would be expected? Does this parameter add anything further? Should it be included?

As we did with the actual D_{KL} , we also included the signed belief update of the Bayesian model into the fMRI GLM (referred to as factor ΔB_t , it was called *directed belief updating* in the supplements of the original manuscript, which has now been corrected). Please see the response to reviewer #1 and Figure R1 for details of signed D_{KL} effects, which demonstrates that neither D_{KL} can be explained by unsigned RPE , nor RPE effects by signed D_{KL} .

The subjective updates that the participants entered are analysed in relation to the ideal updating. That is, we calculated the difference between ideal and actually entered update for every participant on every trial as a regressor in an additional analysis (this, as suggested by reviewer #3, is now included in the main manuscript). In short, this analysis revealed effects in ventral and dorsal striatum, FPC, aMCC, and the a midbrain region overlapping with the VTA/SN whenever participants updated more Bayesian / ideal. The overlap with the D_{KL} contrast itself further strengthens the interpretation that these regions facilitate belief formation.

5) *The explicit instruction to participants is to sample/explore both urns to learn about good or bad long-term outcomes but also to prioritize selecting urns with long-term good consequences to maximize payouts. Because the positive immediate feedback sums were greater for good relative to bad urns, the finding that 153 good and 87 bad urns were sampled on average cannot be interpreted as evidence of learning the long-term outcome associations for each urn. This bias toward selecting good urns could have occurred because good urns on average payout +1.5 points in the immediate/short-term, whereas bad urns on average payout -1.5 points per choice. In this way, the RL model will be slightly biased to favour urns associated with good relative to bad long-term outcomes.*

We agree with the reviewer that, given very long sampling, an RL learner will arrive at an expected value that reflects the true long-term outcome in case the learning rate is very low. This, however, would still be learning. More importantly, an RL learner cannot explain incongruent updating in the correct direction, which participants show. Additionally, we base the conclusion that participants show hallmarks of Bayesian learning in their beliefs not on these overall descriptive statistics, but on a formal model comparison (see Methods p. 17). This indicated overwhelming evidence for a better model fit of the Bayesian, compared to the RL model in explaining participants' beliefs. We furthermore demonstrate in the revised manuscript that beliefs explain choices in the two-urn task (see Fig. S2 or response to reviewer #2 above). Additionally, please consider the very large difference between expected value (± 1.5 points) and each payout per trial (up to 60 points), which makes it very difficult to determine if an urn is *good* or *bad* purely based on payouts. We, however, fully agree that an interesting future study could investigate if participants are actually able to choose the *good* urn more frequently if they are not provided with the event distributions explicitly.

6) *'On average, subjects earned 174 ± 13 points in the task (range: 80 - 280), out of which 67 (± 14) were earned via exploitation within blocks, and 107 (± 4) were earned by correctly estimating urns' long-term valences during blocks as bonus payout.'*

This indicates that participants earned more from bonus questions (potentially 10 per block) at the end of each block than from maximizing points throughout by choosing the urn with good long-term consequences (potential 20 per block). Because the urn with good long-term consequences at times is associated with negative immediate feedback sums, it's unclear how to interpret the number of points earned from participants' sampling behaviour. What was the total possible points that could be earned if participants performed in an ideal way, always sampling from the urn that was associated with positive long-term consequences? This will be difficult to determine because participants need to sample both urns at least on a few trials to determine the long-term consequence associated with each. How did sampling behaviour differ for blocks in which both urns were associated with good or with bad long-term outcomes, relative to blocks in which one urn was good and the other was bad in the long-term? Differences in sampling behaviour between these blocks could potentially provide some evidence of an internal model developing. No difference in sampling between these blocks would favour the view that these findings provide evidence for brain regions that underlie learning versus those that enact decisions.

It is indeed difficult to determine what the ideal strategy for choices in the task would be, which is why we focus on belief formation – for which ideal values can be derived from the Bayesian model. Theoretically, had participants always only sampled the *good* urn and ignored the *bad* urn, they would have gained 300 points during the blocks, and 60 points by correctly estimating the *good* urn and 0 for avoiding the gamble bonus question for the *bad* urn. However, for obvious reasons, this strategy cannot be employed because participants cannot know an urn's true valence from the start.

As suggested by the reviewer, we investigated if sampling differed between good-bad (GB) and good-good (GG) or bad-bad (BB) blocks. To test sampling, we compare the number of switches from one urn to the other, that is if a participant chose to sample from a different urn or the same one again. We observed no difference in this measure depending on block type. Participants changed responses on 18.7% of trials in

the GB blocks (or on average slightly less than 4 times per block), 18.9% in the GG blocks, and 19.6% in the BB block (GB vs GG $p = .94$; GB vs BB $p = .70$).

We acknowledge that we may lack power to detect a difference here, therefore we pooled all the data from the original study, the control behavioural experiment (although with a different task timing that was not optimised for fMRI but should be otherwise comparable), and the control fMRI study together, which results in a sample of 58 participants. Again, we observed no difference in sampling (GB vs GG $p = .79$; GB vs BB $p = .38$). These data are therefore in accordance with assuming that participants actually formed beliefs about the long-term valence of each urn, as is also confirmed by the replication study as explicated above.

5) At odds with the notion that greater DS activity was associated with Bayesian Decision/Learning, they found that the lower the difference between coding Dkl and RPE in DS and FPC, the more optimal was performance and the less biased it was by short-term outcomes. Similarly, to follow up on this contrast of contrasts, the authors regressed individual weights of RL versus Bayesian learning model on the striatal gradients from VS to DS for Dkl and RPE. They found the more RPE signals were focussed ventrally and spared the DS, the more RPE biased belief updates.

Could these findings have arisen from comparing cases in which the DKL and RPE were more similar to one another, relative to cases in which they differed to a greater extent? Was this possibility investigated and excluded? Because trial order was not predetermined and participants' urn selections were idiosyncratic, Bayesian models and RL models will differ for each participant and the difference between these models will also vary.

The informational content of every trial (congruent, incongruent, non-informative) was predetermined. The similarity between RL and Bayesian model almost exclusively depends on the number of incongruent events (in which the RL model updates in accordance with the payout sign, and the Bayesian learner against it). Therefore, model similarity is relatively independent from participants' choices. Furthermore, similarity between regressors in the GLM is unlikely to explain correlation effects across participants, because all shared variance in multiple regression is attributed to the error term and, therefore, individual effects are sure to represent unique variance explained by each regressor. Finally, we tested if the correlation between D_{KL} and RPE regressors displayed any association to the behavioural measure of bias on belief updating. There was no significant association between these measures ($r^2 = 0.029$, $p = .42$). Thus, an explanation of the results based on differential similarity of the models between participants appears very unlikely.

6) One subject was removed from the analysis for not understanding the task instructions. How was this determined? Was this participant removed before analyses were attempted?

This participant was never included into any attempted analysis and was excluded because she consistently pressed the left button during most blocks, indicating that she wrongly attributed outcomes to positions instead of urns, or was simply not motivated to participate. She additionally did not correctly fill out the additional provided questionnaires (she left pages blank). We concluded that this participant was most likely not interested in partaking in the study, but may have been mostly interested in the reimbursement provided. This has been clarified in the revised manuscript on p. 14.

7) There are some unreferenced statements, conclusions that are not well supported by data, or that extend findings too broadly.

We checked the revised manuscript carefully to avoid these problems and believe that due to the helpful comments of this and the other three reviewers, we were able to fully address these concerns.

Minor criticisms:

1) The authors claim that rodents' are unable to simultaneously learn long-term outcomes and disregard immediate rewards. This is presented as a motivation for performing the present imaging study. This statement is not backed up with references. Are there empirical demonstrations? Further, this claim would be at odds with their interpretations that learning from long-term outcomes is mediated by DS, IPS, PFC whereas learning from immediate rewards is mediated by VS, brain regions that have homology in rodents. We removed this statement from the revised manuscript.

2) *'Finally, activity in dopaminergic midbrain regions likely including the ventral tegmental area (Fig. S5) covaried with ideal updating; thus,...'*

This is the first time that authors mention VTA, not described in the results section or even in the figure that is referenced. Because of their proximity, VTA and adjacent SNc activations in the brainstem cannot be differentiated. Authors declare that this activity is most associated with ideal long-term model updating (i.e., learning of long-term consequences), but an equally apt interpretation is that this region correlates with ideal decision making with relation to the conditional probabilities. SNc nearly exclusively innervates DS, the brain region that was shown to most associate with Bayesian Model Learning (or equally Bayesian model Decision making) in this study. Further a large literature ascribes decision making/response selection with DS to provide context. Finally, VS, which is VTA-innervated, was most associated with RL model (RPE) and to a much lesser extent the Bayesian Learner model, to which the ideal updating refers.

We thank the reviewer for this insightful comment and agree that 3T fMRI is not suited to differentiate subnuclei within the midbrain. We furthermore agree that SNc would be a more plausible candidate region causing the observed effect given its tight interconnection with the DS.

Overall, we toned down the interpretation of the effects, yet we feel that they are noteworthy, given that the midbrain effect is not only seen for the „trial-by-trial Bayesianness“ regression, but also for a main effect of D_{KL} . In the revised manuscript, we report these findings along with the results for single-trial Bayesianness which we included, as suggested by reviewer #3, into the main manuscript on p. 8.

3) *'This appears compatible with previous findings that activity in dopaminergic midbrain regions increases with task demands^{26,27}: it could be speculated that increased midbrain activity could support ventral-to-dorsal information transfer and integration within the striatum and increase gain in cognitive control areas.'* *This was not demonstrated by the results. Correlations between midbrain regions, VS versus DS, and behavioural change were not investigated.*

We have toned this down and state this more explicitly as a speculative explanation in the results as well as in a short part of the discussion (see response to minor point 2 above).

4) *'Perhaps this could help to explain the success of mindfulness-based therapies in various psychiatric fields including the therapy of addiction'*

This extends the findings too broadly and is an unnecessary statement.

We have removed this sentence from the manuscript.

Reviewers' comments:

Editorial Note: As Reviewer #1 was unable to provide a timely re-review, we asked the other reviewers to ensure that the authors' responses to his/her concerns were adequate.

Reviewer #2 (Remarks to the Author):

The authors have addressed all my comments. Excellent that they have performed a replication sample.

Reviewer #3 (Remarks to the Author):

1-7) I am happy with the authors changes

8) I am happy for this analysis not to be included, but thank the authors for telling me about the results!

All minor comments have been addressed to my satisfaction.

I congratulate the authors on a great paper and hope my previous comments were helpful.

Reviewer #4 (Remarks to the Author):

First, I congratulate the authors on a very thorough revision, including conducting a replication/extension experiment to address a concern with the original design. The experiment is now better explained. The questions that are pursued in this manuscript are of great importance and interest. The methodology is novel and clever. I have a few remaining concerns, however, detailed below.

1) The abstract and the introduction occasionally lose focus of what the authors are purportedly investigating in this manuscript. The authors vacillate between the topic of competing influences on decision making and learning over the short and long terms in model-free versus model-based fashion. This is despite the facts that they lay out an experiment aimed at investigating different forms of learning, and interpret the results of their experiments as evidence of dissociated brain regions mediating learning from immediate rewards/punishments versus from information collected over many trials to achieve a model in the long-term. The topic under study should be consistently presented. Further, framing the current experiment as relating to decision making and learning intermittently seems to equate short-term and long-term consequences of decisions with learning through immediate outcomes (i.e., model-free) and learning through inferences drawn over a number of experiences. These concepts in decision making and learning are not equal and do not map onto one another.

The authors even cast their results in these conflicting manners on a few occasions. I refer the authors to the section entitled: Representation of model-free signals in regions that reflect model-based learning predicts learning performance. In this section, they begin by assigning to significantly activated brain regions, the cognitive process of making decisions

on the basis of the Bayesian model and overcoming biases that arise from model-free learning. They explain that this is not unlike the real-world example of choosing exercise over consumption of fast-food. Subsequently in this section, however, the authors attribute more simultaneous representation of DKL and RPE in DS to superior learning of long-term beliefs about future outcomes. This is confusing.

2) The new experiment is important in confirming that participants are learning long-term associations over a number of trials incorporating information gained from immediate feedback using a pre-defined structure to form a model for categorizing urns as good or bad. Neither the original nor the new experiment, however, can clearly confirm that brain regions whose activation change across the experiment mediate learning. That is, brain regions implicated in acting on learning will also undergo changes as these associations are acquired. The authors attempt to address this issue in the extension experiment by having participants explicitly state the valence of an urn on only every 3rd trial. In this experiment, they include as a regressor whether a belief was explicitly prompted or not after the immediate feedback. An interaction between this regressor and DKL (i.e., the model of Bayesian learning over the long-term) with respect to the activation of a brain region would suggest that beliefs are only formed when participants need to make a decision about urn valence. This would favour the view that this brain region was coding information about the model in the service of decision making about urn valence. On the other hand, if the interaction is not significant, this would indicate that belief updates and their neuronal correlates were not confined to prompt trials. The authors suggest that the absence of an interaction would favour the view that the brain region was in fact mediating learning and not decision making based on new learning. In the follow up experiment, L IPS, dlPFC, and pmFC exhibited a significant interaction (at strong or weaker thresholds) suggesting belief updating and change in neuronal activation on prompt relative to non-prompt trials only or to a greater extent. Given their hypotheses and the design of this extension experiment, this interaction favours the view that the updating in activation of these regions reflected improved decision making. This is not clearly expressed in the discussion section. In contrast, the lack of interaction for the DS and PFC is interpreted by the authors as confirmation that these brain regions are mediating the learning of the long-term belief in a model-based fashion. There are a number of problems with this interpretation. It relies on accepting a null hypothesis. The possibility that this contrast was underpowered to find true differences between prompt and non-prompt trials needs to be explored further. Also, in separate analyses with prompt versus non-prompt trials, did similar patterns of correlation with DKL emerge in these brain regions? This is an important contrast to perform to bolster the authors claims. Regardless of these findings, however, their discussion of these results must be qualified by the potential of an alternative interpretation. It is overstating the case that the results of the extension experiment rule out a potential confound and another explanation for their findings. The discussion in the main text and in the supplementary text must be tempered in this light. Indeed, the authors acknowledge this persistent confound in their Response to Reviewers letter. Brain regions that are downstream from and hence informed by those actually mediating learning will undergo a similar change in activation patterns. Further, in fact, they cannot fully preclude that on non-prompt trials participants were making judgements about the valence in the long-term of each urn. Participants would be motivated to perform these decisions on every trial in fact, even if they are not asked to

explicitly offer their judgment, as this information collected on every trial will improve the accuracy of their end-of-block judgments which leads to bonus points.

The following for example overstates what the extension experiment accomplishes:

To rigorously rule out that the neural correlates we observed were induced by prompting beliefs, we conducted a control experiment in which belief prompts were presented only on every third trial and participants had to memorize positions of the belief bar. This ensures that participants formed an internal representation of the belief and avoids a possible confound between learning and decision-making 25 (Supplemental Discussion and Fig. S6). This confirmed that these neural correlates reflect learning via inference and can be separated from short-term, model-free reward processing.

Overall, the potential that signals in brain regions could reflect processes related to decisions and performance as opposed to learning per se must be considered as an alternative interpretation throughout the manuscript where this cannot be eliminated. This applies to discussions regarding optimality of learning/responding or incongruence of model-free versus model-based.

Response Letter

We thank the reviewers for their positive assessments of our revision. Please see our responses below (in blue print for clarity).

Reviewer #2 (Remarks to the Author):

The authors have addressed all my comments. Excellent that they have performed a replication sample. We thank the reviewer for this assessment and the helpful suggestions and comments which certainly strengthened the manuscript.

Reviewer #3 (Remarks to the Author):

1-7) I am happy with the authors changes

8) I am happy for this analysis not to be included, but thank the authors for telling me about the results! All minor comments have been addressed to my satisfaction.

I congratulate the authors on a great paper and hope my previous comments were helpful.

As suggested by the editor, we included a short summary of the stimulus-locked data in the supplements. We thank the reviewer very much for the insightful comments and we are convinced that the manuscript benefitted considerably for the suggested changes.

Reviewer #4 (Remarks to the Author):

First, I congratulate the authors on a very thorough revision, including conducting a replication/extension experiment to address a concern with the original design. The experiment is now better explained. The questions that are pursued in this manuscript are of great importance and interest. The methodology is novel and clever. I have a few remaining concerns, however, detailed below.

We are happy with the reviewer's assessment of the revision. We feel that the reviewer's critical view and insightful comments regarding the differentiability of learning and decision making mechanisms have led to a much improved manuscript. We respond to the remaining concerns below.

1) The abstract and the introduction occasionally lose focus of what the authors are purportedly investigating in this manuscript. The authors vacillate between the topic of competing influences on decision making and learning over the short and long terms in model-free versus model-based fashion. This is despite the facts that they lay out an experiment aimed at investigating different forms of learning, and interpret the results of their experiments as evidence of dissociated brain regions mediating learning from immediate rewards/punishments versus from information collected over many trials to achieve a model in the long-term. The topic under study should be consistently presented. Further, framing the current experiment as relating to decision making and learning intermittently seems to equate short-term and long-term consequences of decisions with learning through immediate outcomes (i.e., model-free) and learning through inferences drawn over a number of experiences.

These concepts in decision making and learning are not equal and do not map onto one another.

The authors even cast their results in these conflicting manners on a few occasions. I refer the authors to the section entitled: Representation of model-free signals in regions that reflect model-based learning predicts learning performance. In this section, they begin by assigning to significantly activated brain regions, the cognitive process of making decisions on the basis of the Bayesian model and overcoming biases that arise from model-free learning. They explain that this is not unlike the real-world example of choosing exercise over consumption of fast-food. Subsequently in this section, however, the authors attribute more

simultaneous representation of DKL and RPE in DS to superior learning of long-term beliefs about future outcomes. This is confusing.

We thank the reviewer for the very thorough and detailed corrections. We have corrected these formulations and describe the results in the framework of learning, while acknowledging that this is tightly intertwined with application of learned value in decision making in more detail in the discussion.

2) The new experiment is important in confirming that participants are learning long-term associations over a number of trials incorporating information gained from immediate feedback using a pre-defined structure to form a model for categorizing urns as good or bad. Neither the original nor the new experiment, however, can clearly confirm that brain regions whose activation change across the experiment mediate learning. That is, brain regions implicated in acting on learning will also undergo changes as these associations are acquired. The authors attempt to address this issue in the extension experiment by having participants explicitly state the valence of an urn on only every 3rd trial. In this experiment, they include as a regressor whether a belief was explicitly prompted or not after the immediate feedback. An interaction between this regressor and DKL (i.e., the model of Bayesian learning over the long-term) with respect to the activation of a brain region would suggest that beliefs are only formed when participants need to make a decision about urn valence. This would favour the view that this brain region was coding information about the model in the service of decision making about urn valence. On the other hand, if the interaction is not significant, this would indicate that belief updates and their neuronal correlates were not confined to prompt trials. The authors suggest that the absence of an interaction would favour the view that the brain region was in fact mediating learning and not decision making based on new learning. In the follow up experiment, L IPS, dlPFC, and pmPFC exhibited a significant interaction (at strong or weaker thresholds) suggesting belief updating and change in neuronal activation on prompt relative to non-prompt trials only or to a greater extent. Given their hypotheses and the design of this extension experiment, this interaction favours the view that the updating in activation of these regions reflected improved decision making. This is not clearly expressed in the discussion section. In contrast, the lack of interaction for the DS and PFC is interpreted by the authors as confirmation that these brain regions are mediating the learning of the long-term belief in a model-based fashion. There are a number of problems with this interpretation. It relies on accepting a null hypothesis. The possibility that this contrast was underpowered to find true differences between prompt and non-prompt trials needs to be explored further. Also, in separate analyses with prompt versus non-prompt trials, did similar patterns of correlation with DKL emerge in these brain regions? This is an important contrast to perform to bolster the authors claims. Regardless of these findings, however, their discussion of these results must be qualified by the potential of an alternative interpretation. It is overstating the case that the results of the extension experiment rule out a potential confound and another explanation for their findings. The discussion in the main text and in the supplementary text must be tempered in this light. Indeed, the authors acknowledge this persistent confound in their Response to Reviewers letter. Brain regions that are downstream from and hence informed by those actually mediating learning will undergo a similar change in activation patterns. Further, in fact, they cannot fully preclude that on non-prompt trials participants were making judgements about the valence in the long-term of each urn. Participants would be motivated to perform these decisions on every trial in fact, even if they are not asked to explicitly offer their judgment, as this information collected on every trial will improve the accuracy of their end-of-block judgments which leads to bonus points.

The following for example overstates what the extension experiment accomplishes:

To rigorously rule out that the neural correlates we observed were induced by prompting beliefs, we conducted a control experiment in which belief prompts were presented only on every third trial and participants had to memorize positions of the belief bar. This ensures that participants formed an internal representation of the belief and avoids a possible confound between learning and decision-making 25 (Supplemental Discussion and Fig. S6). This confirmed that these neural correlates reflect learning via inference and can be separated from short-term, model-free reward processing.

Overall, the potential that signals in brain regions could reflect processes related to decisions and performance as opposed to learning per se must be considered as an alternative interpretation throughout

the manuscript where this cannot be eliminated. This applies to discussions regarding optimality of learning/responding or incongruence of model-free versus model-based.

We agree with the reviewer and the editor that reporting only the interaction null-findings is not sufficient evidence to favour one interpretation (learning) over the other (decision making). We apologize for this imprecision and, as suggested, now include the main contrast of D_{KL} exclusively for trials not followed by a belief prompt. Thus, if results here remained significant, belief updating is independent of belief prompting, ruling out the confound that the prompt caused the observed effects. We found that all results clearly remained significant (Figure S7c). The absence of an interaction (at the same threshold applied to the other analyses) furthermore clearly hints at similar processing in both conditions. This provides strong evidence in favour of an internal model forming that participants use to base their decisions on, which we term learning.

The reviewer repeatedly pointed out the importance of disentangling learning and decision making processes. While it may seem relatively straightforward to study decision making without significant learning (by externally providing values for each decision), it appears extremely difficult to minimize decision making processes in learning experiments: to objectively test learning performance, decisions are inevitable to read out the learned values, beliefs etc. Acknowledging this, we followed the reviewer's suggestion and toned down all statements relating to the relative degree of evidence in favour of either learning or decision making based on new learned beliefs. In addition, by introducing no-prompt trials in the replication study, we found a way of separating learning and decision making as much as possible (both, temporally as well as with respect to the fact that on no-prompt trials no decisions on how to move the prompt are necessary). In our view, the fact that neural correlates of Bayesian model-updating are a) fully present when participants do not need to decide about choosing an update and b) not significantly different between when they do have to do so, despite the possibility of a lack of power, favours the explanation that these regions mediate inference-based learning. We are thankful for the reviewer's critical comments which clearly have sharpened the focus of the manuscript. We furthermore present the task to more precisely disentangle learning from decision making as a fruitful challenge for further studies in the discussion.

REVIEWERS' COMMENTS:

Reviewer #4 (Remarks to the Author):

Thank you for this careful revision.